# REDUCE, REUSE, RECYCLE: COMPOSITIONAL GENERATION WITH ENERGY-BASED DIFFUSION MODELS AND MCMC

## ABSTRACT

Since their introduction, diffusion models have quickly become the prevailing approach to generative modeling in many domains. They can be interpreted as learning the gradients of a time-varying sequence of log-probability density functions. This interpretation has motivated classifier-based and classifier-free guidance as methods for post-hoc control of diffusion models. In this work, we build upon these ideas using the score-based interpretation of diffusion models, and explore alternative ways to condition, modify, and reuse diffusion models for tasks involving compositional generation and guidance. In particular, we investigate why certain types of composition fail using current techniques and present a number of solutions. We conclude that the sampler (not the model) is responsible for this failure and propose new samplers, inspired by MCMC, which enable successful compositional generation. Further, we propose an energy-based parameterization of diffusion models which enables the use of new compositional operators and more sophisticated, Metropolis-corrected samplers. Intriguingly we find these samplers lead to notable improvements in compositional generation across a wide variety of problems such as classifier-guided ImageNet modeling and compositional text-to-image generation.

## 1 INTRODUCTION

In recent years, tremendous progress has been made in generative modeling across a variety of domains (Brown et al., 2020; Brock et al., 2018; Ho et al., 2020). These models now serve as powerful priors for downstream applications such as code generation (Li et al., 2022), text-to-image generation (Saharia et al., 2022), question-answering (Brown et al., 2020) and many more. However, to fit this complex data, generative models have grown inexorably larger (requiring 10's or even 100's of billions of parameters) (Kaplan et al., 2020) and require datasets containing non-negligible fractions of the entire internet, making it costly and difficult to train and or finetune such models. Despite this, some of the most compelling applications of large generative models do not rely on finetuning. For example, prompting (Brown et al., 2020) has been a successful strategy to selectively extract insights from large models. In this paper, we explore an alternative to finetuning and prompting, through which we may repurpose the underlying prior learned by generative models for downstream tasks.

Diffusion Models (Sohl-Dickstein et al., 2015; Song & Ermon, 2019; Ho et al., 2020) are a recently popular approach to generative modeling which have demonstrated a favorable combination of scalability, sample quality, and log-likelihood. A key feature of diffusion models is the ability for their sampling to be "guided" after training. This involves combining the pre-trained Diffusion Model $p_\theta(x)$ with a predictive model $p_\theta(y|x)$ to generate samples from $p_\theta(x|y)$. This predictive model can be either explicitly defined (such as a pre-trained classifier) (Sohl-Dickstein et al., 2015; Dhariwal & Nichol, 2021) or an implicit predictive model defined through the combination of a conditional and unconditional generative model (Ho & Salimans, 2022). These forms of conditioning are particularly appealing (especially the former) as they allow us to reuse pre-trained generative models for many downstream applications, beyond those considered at training time.

These conditioning methods are a form of model composition, i.e. combining probabilistic models together to create new models. Compositional models have a long history back to early work on Mixtures-Of-Experts (Jacobs et al., 1991) and Product-Of-Experts models (Hinton, 2002; Mayraz &

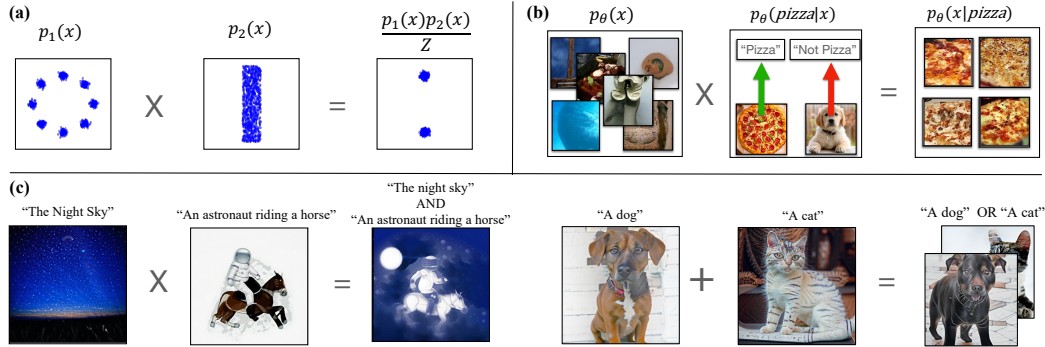

Figure 1: **Creating new models through composition.** Simple operators enable diffusion models to be composed without retraining in settings such **(a)** products, **(b)** classifier conditioning, **(c)** compositional text-to-image generation with a product (left) and a mixture (right). All samples generated by trained models.

Hinton, 2000). Here, many simple models or predictors were combined to increase their capacity. Much of this early work on model composition was done in the context of Energy-Based Models (Hinton, 2002), an alternative class of generative model which bears many similarities to diffusion models.

In this work, we explore the ways that diffusion models can be reused and composed with one-another. First, we introduce a set of methods which allow pre-trained diffusion models to be composed, with one-another and with other models, to create new models without retraining. Second, we illustrate how existing methods for composing diffusion models are not fully correct, and propose a remedy to these issues with MCMC-derived sampling. Next, we propose the use of an energy-based parameterization for diffusion models, where the unnormalized density of each reverse diffusion distribution is explicitly modeled. We illustrate how this parameterization enables both additional ways to compose diffusion models, as well as the use of more powerful Metropolis-adjusted MCMC samplers. Finally, we demonstrate the effectiveness of our approach in settings from 2D data to high-resolution text-to-image generation. An illustration of our domains can be found in Figure 1.

## 2 BACKGROUND

### 2.1 DIFFUSION MODELS

Diffusion models seek to model a data distribution $q(x_0)$. We augment this distribution with auxiliary variables $\{x_t\}_{t=1}^T$ defining a Gaussian diffusion $q(x_0, \ldots, x_T) = q(x_0)q(x_1|x_0)\ldots q(x_T|x_{T-1})$ where each transition is defined $q(x_t|x_{t-1}) = \mathcal{N}(x_t; \sqrt{1-\beta_t}x_{t-1}, \beta_t I)$ for some $0 < \beta_t \leq 1$. This transition first scales down $x_{t-1}$ by $\sqrt{1-\beta_t}$ and then adds Gaussian noise of variance $\beta_t$. For large enough $T$, we will have $q(x_T) \approx \mathcal{N}(0, I)$.

Our model takes the form $p_\theta(x_{t-1}|x_t)$ and seeks to learn the reverse distribution of $q(x_t|x_{t-1})$ which seeks to denoise $x_t$ to $x_{t-1}$. In the limit of small $\beta_t$ this reversal becomes Gaussian (Sohl-Dickstein et al., 2015) so we parameterize our model $p_\theta(x_{t-1}|x_t) = \mathcal{N}(x_{t-1}; \mu_\theta(x_t, t), \tilde{\beta}_t I)$ with:

$$\mu_\theta(x_t, t) = \frac{1}{\sqrt{\alpha_t}}\left(x_t - \frac{\beta_t}{\sqrt{1-\bar{\alpha}_t}}\epsilon_\theta(x_t, t)\right). \tag{1}$$

where $\epsilon_\theta(x_t, t)$ is a neural network, and $\alpha_t, \bar{\alpha}_t, \tilde{\beta}_t$ are functions of $\{\beta_t\}_{t=1}^T$.

A useful feature of the diffusion process $q$ is that we can analytically derive any time marginal $q(x_t|x_0) = \mathcal{N}(x_t; \sqrt{1-\sigma_t^2}x_0, \sigma_t^2 I)$ where again $\sigma_t$ is a function of $\{\beta_t\}_{t=1}^T$. We can sample $x_t$ from this distribution using reparameterization, i.e $x_t(x_0, \epsilon) = \sqrt{1-\sigma_t^2}x_0 + \sigma_t\epsilon$ where $\epsilon \sim \mathcal{N}(0, I)$. Exploiting this, diffusion models are typically trained with the loss

$$\mathcal{L}(\theta) = \sum_{t=1}^T \mathcal{L}_t(\theta), \qquad \mathcal{L}_t(\theta) = \mathbb{E}_{q(x_0)\mathcal{N}(\epsilon;0,I)}\left[||\epsilon - \epsilon_\theta(x_t(x_0, \epsilon), t)||^2\right]. \tag{2}$$

Once $\epsilon_\theta(x, t)$ is trained, we recover $\mu_\theta(x, t)$ with Equation 1 to parameterize $p_\theta(x_{t-1}|x_t)$ and perform ancestral sampling (also known as the reverse process) to reverse the diffusion, i.e sample $x_T \sim \mathcal{N}(0, I)$, then for $t = T - 1 \to 1$, sample $x_{t-1} \sim p_\theta(x_{t-1}|x_t)$. A more detailed description can be found in Appendix B.

## 2.2 Energy-Based Models and MCMC Sampling

Energy-Based Models (EBMs) are a class of probabilistic model which parameterize a distribution as $p_\theta(x) = \frac{e^{f_\theta(x)}}{Z(\theta)}$ where the normalizing constant $Z(\theta) = \int e^{f_\theta(x)} dx$ is not modeled. Choosing not to model this quantity gives the model much more flexibility but comes with considerable limitations. We can no longer efficiently compute likelihoods or draw samples from the model. This complicates training, as most generative models are trained by maximizing likelihood.

One popular method for EBM training is denoising score matching. This approach minimizes the Fisher Divergence[1] between the model and a Gaussian-smoothed version of the data distribution $q_\sigma(x) = \int q(x') \mathcal{N}(x; x', \sigma^2 I) dx'$ by minimizing the following objective

$$\mathcal{J}_\sigma(\theta) = \mathbb{E}_{q(x)\mathcal{N}(\epsilon;0,I)} \left[ \left|\left| \tfrac{\epsilon}{\sigma} + \nabla_x f_\theta(x + \sigma\epsilon) \right|\right|^2 \right]. \tag{3}$$

When minimized, this ensures that $e^{f_\theta(x)} \propto q_\sigma(x)$ and therefore $\nabla_x f_\theta(x) = \nabla_x \log q_\sigma(x)$. To estimate likelihoods or sample from our model, we must rely on approximate methods, such as MCMC sampling or numerical ODE integration. MCMC works by simulating a Markov chain beginning at $x_0 \sim p(x_0)$ and using a transition distribution $x_t \sim k(x_t|x_{t-1})$. If $k(\cdot|\cdot)$ has certain properties, namely invariance w.r.t. the target and ergodicity, then as $t \to \infty$, $x_t$ converges to a sample from our target distribution.

Perhaps the most popular MCMC sampling algorithm for EBMs is Unadjusted Langevin Dynamics (ULA) (Roberts & Tweedie, 1996; Du & Mordatch, 2019; Nijkamp et al., 2020) which is defined by

$$k(x_t|x_{t-1}) = \mathcal{N}\left(x_t; x_{t-1} + \tfrac{\sigma^2}{2} \nabla_x f_\theta(x_{t-1}), \sigma^2 I\right). \tag{4}$$

This resembles a step of gradient ascent (with step-size $\frac{\sigma^2}{2}$) with added Gaussian noise of variance $\sigma^2$. This transition is based on a discretization of the Langevin SDE. In the limit of infinitesimally small $\sigma$ this approach will draw exact samples. To handle the error accrued when using larger step sizes, a Metropolis correction can be added giving the Metropolis-Adjusted-Langevin-Algorithm (MALA) (Besag, 1994). With Metropolis correction, we first generate a proposed update $\hat{x} \sim k(x|x_{t-1})$, then with probability $\min\left(1, \frac{e^{f_\theta(\hat{x})}}{e^{f_\theta(x_{t-1})}} \frac{k(x_{t-1}|\hat{x})}{k(\hat{x}|x_{t-1})}\right)$ we set $x_t = \hat{x}$, otherwise $x_t = x_{t-1}$.

Hamiltonian Monte Carlo (HMC) (Duane et al., 1987; Neal, 1996) is a more advanced MCMC sampling method which augments the state-space with auxiliary momentum variables and numerically integrates energy-conserving Hamiltonian dynamics to advance the sampler. HMC is typically applied with a Metropolis correction, but an approximate variant can be used without it (U-HMC) (Geffner & Domke, 2021). See Appendix C.1 for details of HMC variants we use.

## 2.3 Relationship Between Diffusion Models and EBMs

Diffusion models and EBMs are closely related. For instance, Song & Ermon (2019) uses an EBM perspective to propose a close cousin to diffusion models. We can see from inspection that the training objective of diffusion models is identical (up to a constant) to the denoising score matching objective

$$\sigma_t^2 \mathcal{J}_{\sigma_t}(\theta) = \mathbb{E}_{q(x)\mathcal{N}(\epsilon;0,I)} \left[ ||\epsilon + \sigma_t \nabla_x f_\theta(x + \sigma_t\epsilon)||^2 \right] = \mathcal{L}_t(\theta) \tag{5}$$

where we have replaced $\epsilon_\theta(x, t)$ with $-\sigma_t \nabla_x f_\theta(x + \sigma_t\epsilon)$. Thus by training $\epsilon_\theta(x, t)$ to minimize Equation 2, we can recover the diffused data distribution score with $\nabla_x \log q_\sigma(x) \approx -\frac{\epsilon_\theta(x,t)}{\sigma_t}$. From this, we can define $\epsilon_\theta(x, t) = \nabla_x f_\theta(x, t)$ (the derivative of an explicitly defined scalar function) to learn a noise-conditional potential function $f_\theta(x, t)$. We later demonstrate the benefits of this in two ways; it enables the use of more sophisticated sampling algorithms and more forms of composition.

## 2.4 Controllable Generation

It may be convenient to train a model of $p(x)$ where $x$ is, say, the distribution of all images, but in practice we often want to generate samples from $p(x|y)$ where $y$ is some attribute, label, or feature. This can be accomplished within the framework of diffusion models by introducing a learned

---

[1]The Fisher divergence is defined: $\mathbf{F}(p||q) = \mathbb{E}_p \left[ ||\nabla_x \log p(x) - \nabla_x \log q(x)||^2 \right]$.

predictive model $p_\theta(y|x;t)$, i.e a time-conditional model of the distribution of some feature $y$ given $x$. We can then exploit Bayes' rule to notice that (for $\lambda = 1$),

$$\nabla_x \log p_\theta(x|y;t) = \nabla_x \log p_\theta(x;t) + \lambda \nabla_x \log p_\theta(y|x;t). \tag{6}$$

In practice, when using the right side of Equation 6 for sampling, it is beneficial to increase the 'guidance scale' $\lambda$ to be $> 1$ (Dhariwal & Nichol, 2021). Thus, we can re-purpose the unconditional diffusion model and turn it into a conditional model.

If instead of a classifier, we have a both an unconditional diffusion model $\nabla_x \log p_\theta(x;t)$ and a conditional diffusion model $\nabla_x \log p_\theta(x|y;t)$, we can again utilize Bayes' rule to derive an implicit predictive model's gradients

$$\nabla_x \log p_\theta(y|x;t) = \nabla_x \log p_\theta(x|y;t) - \nabla_x \log p_\theta(x;t) \tag{7}$$

which can be used to replace the explicit model in Equation 6, giving what is known as classifier-free guidance (Ho & Salimans, 2022). This method has led to incredible performance, but comes at a cost to modularity. This contrasts with the classifier-guidance setting, where we only need to train a single (costly) generative model. We can then attach any predictive model we would like to for conditioning. This is beneficial as it is often much easier and cheaper to train predictive models than a flexible generative model. In the classifier-free setting, we must know exactly which $y$ we would like to condition on, and incorporate these labels into model training. In both guidance settings, we use our (possibly implicit) predictive model to modify the learned score of our model. We then perform diffusion model sampling as we would in the unconditional setting. We will see later that even in toy settings, this is often not the optimal thing to do.

## 3 COMPOSITIONAL GENERATION BEYOND GUIDANCE

Most work on conditional diffusion models has come in the form of classifier or classifier-free guidance, but these are far from the only ways we can compose distributions to obtain new models. These ideas have been studied primarily in the context of EBMs because most compositional operators leave the resulting distribution unnormalized. We outline various options below.

**Products:** We can take a product of $N$ distributions and re-normalize to create a new distribution, roughly equivalent to the "intersection" of the composite distributions,

$$q^{\text{prod}}(x) = \frac{1}{Z} \prod_{i=1}^{N} q^i(x), \qquad Z = \int \prod_{i=1}^{N} q^i(x)dx. \tag{8}$$

Regions of high probability under $q^{\text{prod}}(x)$ will typically have high probability under *all* $q^i(x)$. A simple product model can be seen in Figure 2. These ideas were initially proposed to increase the capacity of weaker models by allowing individual "experts" to model specific features in the input (Hinton, 2002), and were recently demonstrated at scale in the image domain using Deep Energy-Based Models (Du et al., 2020a).

The approaches to guidance discussed in Section 2.4 define product models with only two experts. The first models the relative density of the input data and the second models the conditional probability of $y$. Combining these by a product models likely inputs which have the desired property $y$. This form of composition has become popular for diffusion models since they do not directly model the probability, but instead the gradient of the log-probability which can also be composed in this way.

**Mixtures:** Complementary to the product or intersection is the mixture or union of multiple distributions. We can combine $N$ distributions through a mixture to create a new distribution equivalent to the *union* of the concepts captured in each distribution

$$q^{\text{mix}}(x) = \frac{1}{N} \sum_{i=1}^{N} q^i(x) \tag{9}$$

where regions of high probability consist of regions of high probability under *any* $q^i(x)$. We cannot compose score-functions to define mixtures (unlike products). Instead, we need a model which specifies probability. Generating from mixtures of energy based models requires knowing the ratio of normalizers between the models. In our experiments, we assume this ratio is 1. A simple compositional mixture model can be seen in Figure 2.

**Negation:** Finally, given two distributions $p_0(x)$ and $p_1(x)$, we can explicitly invert the density of $p_1(x)$ with respect to $p_0(x)$, which constructs a new distribution which assigns high likelihood to

Figure 2: **An illustration of product and mixture compositional models, and the improved sampling performance of MCMC in both cases.** Left to right: Component distributions, ground truth composed distribution, reverse diffusion samples, HMC samples. Top: product, bottom: mixture. Reverse diffusion fails to sample from composed models.

points in $p_0(x)$ that are not in $p_1(x)$ (Du et al., 2020a), where $\alpha$ controls the degree we invert $p_1(x)$ (we use $\alpha = 0.5$ in our experiments).

$$q^{\text{neg}}(x) \propto \frac{q^0(x)}{q^1(x)^\alpha}. \tag{10}$$

We can combine negation with our previous operators, in a nested manner to construct complex combinations of distributions (Figure 5).

In section 2.3, we showed how diffusion models can be interpreted as approximating the gradient $\nabla_x \log q(x)$, but do not learn an explicit model of the log-likelihood $\log q(x)$. This means with the standard $\epsilon_\theta(x, t)$-parameterization we can, in theory, utilize product and negation composition, but *not* mixture composition.

# 4 SCALING COMPOSITIONAL GENERATION WITH DIFFUSION MODELS

While highly compositional, EBMs present many challenges. The lack of a normalized probability function makes training, evaluation, and sampling very difficult. Much progress has been made to scale these models (Du & Mordatch, 2019; Nijkamp et al., 2020; Grathwohl et al., 2019; Du et al., 2020b; Grathwohl et al., 2021), but EBMs still lag behind other approaches in terms of efficiency and scalability. In contrast, diffusion models have demonstrated very impressive scalability. Fortuitously, diffusion models have similarities to EBMs, such as their training objective and their score-based interpretation, which makes many forms of composition readily applicable.

Unfortunately, when two diffusion models are composed into, for example, a product model $q^{\text{prod}}(x) \propto q^1(x)q^2(x)$, issues arise if the model which reverses the diffusion uses a score estimate obtained by adding the score estimates of the two models. We see in Figure 2 that composing two models in such a way leads indeed to sub-par samples. This is because to sample from this product distribution using standard reverse diffusion (Song et al., 2021), one would need to compute instead the score of the diffused target product distribution given by

$$\nabla_x \log \tilde{q}_t^{\text{prod}}(x_t) = \nabla_x \log \left( \int dx_0 q^1(x_0)q^2(x_0) \, q(x_t|x_0) \right). \tag{11}$$

For $t > 0$, this quantity is *not* equal to the sum of the scores of the two models which is given by

$$\nabla_x \log q_t^{\text{prod}}(x_t) = \nabla_x \log \left( \int dx_0 q^1(x_0)q(x_t|x_0) \right) + \nabla_x \log \left( \int dx_0 q^2(x_0)q(x_t|x_0) \right). \tag{12}$$

Therefore, plugging the composed score function into the standard ancestral sampling procedure discussed in Section 2.1, which we refer to as "reverse diffusion," does not correspond to sampling from the composed model, and thus reverse diffusion sampling will generate incorrect samples from composed distributions. This effect can be seen in Figure 2, with details in Appendix D.

The score of the distribution $q_t^{\text{prod}}(x_t)$ in Equation 12 is easy to compute, unlike that of $\tilde{q}_t^{\text{prod}}(x_t)$ from Equation 11. In addition, $q_t^{\text{prod}}(x_t)$ describes a sequence of distributions which smoothly interpolate between $q^{\text{prod}}(x)$ at $t = 0$ and $\mathcal{N}(0, I)$ at $t = T$, though this sequence of distributions does not correspond to the distributions that result from the standard forward diffusion process described in Section 2.1, leading the reverse diffusion sampling to generate poor samples. We discuss how we may utilize MCMC samplers, which use our knowledge of $\nabla_x \log q_t^{\text{prod}}(x_t)$, to correctly sample from intermediate distributions $\tilde{q}_t^{\text{prod}}(x_t)$, leading to accurate composed sample generation.

## 4.1 IMPROVING SAMPLING WITH MCMC

In order to sample from $q^{\text{prod}}(x)$ using the combined score function from Equation 12, we can use annealed MCMC sampling, described below in Algorithm 1. This method applies MCMC transition

| Model | Sampler | Product | | | Mixture | | |
|-------|---------|---------|------|-------|----------|------|-------|
| | | RAISE ↑ | LL ↑ | Var ↓ | ln(MMD) ↓ | LL ↑ | Var ↓ |
| Score | Reverse | 1.55 | -6.47 | 0.063 | - | - | - |
| | ULA | 2.37 | 1.79 | 0.026 | - | - | - |
| | U-HMC | **2.52** | **2.40** | **0.021** | - | - | - |
| | Reverse (equal steps) | 2.27 | -2.92 | 0.046 | - | - | - |
| EBM | Reverse | 1.37 | -6.03 | 0.064 | -3.84 | -2.17 | 0.020 |
| | ULA | 2.36 | 1.84 | 0.027 | -4.21 | 0.57 | 0.013 |
| | MALA | 2.64 | 2.73 | 0.013 | -4.38 | 1.29 | 0.008 |
| | U-HMC | 2.63 | 2.45 | 0.022 | **-4.69** | 1.03 | 0.010 |
| | HMC | **2.71** | **2.72** | **0.009** | -4.48 | **1.30** | **0.007** |

Table 1: Quantitative results on 2D composition. **Energy based parameterization enables mixture compositional models, and MCMC sampling leads to better samples from compositional diffusion models.**

kernels to a sequence of distributions which begins with a known, tractable distribution and concludes at our target distribution. Annealed MCMC has a long history enabling sampling from very complex distributions (Neal, 2001; Song & Ermon, 2019).

We explore two types of transition kernels $k_t(\cdot|\cdot)$ based on Langevin Dynamics (Equation 4) and HMC. When using the standard $\epsilon_\theta(x, t)$-parameterization, we do not have access to an explicitly defined energy-function meaning we cannot utilize any MCMC sampler with Metropolis corrections. Thus, we only utilize the ULA and U-HMC samplers described in Section 2.2. These samplers are not exact, but can in practice generate good results. In the next section we detail how Metropolis corrections may be incorporated. Full details of our samplers can be found in Appendix C.1. While continuous time sampling in diffusion models Song et al. (2021) is also referred to as ULA, the MCMC sampling procedure is run across time (and corresponds to the same sampling procedure as discretized diffusion discussed in Section 2.1), as opposed to being used to sample from each intermediate distribution $\tilde{q}_t^{\text{prod}}(x_t)$. Thus applying continuous sampling gives the same issues as reverse diffusion sampling.

---

**Algorithm 1** Annealed MCMC

**Input:** Transition kernels $k_t(\cdot|\cdot)$, Initial distribution $p_T(\cdot)$, Number of steps $N$
$x_T \sim p_T(x)$     # Initialize.
**for** $t = T, \ldots, 0$ **do**
    **for** $i = 1, \ldots, N$ **do**
        $x_t \sim k_t(\cdot|x_t)$
    **end for**
    $x_{t-1} = x_t$
**end for**
**return** $x_0$

---

We can see again in Figure 2 that applying this MCMC sampling procedure allows samples from the composed distribution to be faithfully generated with no modification to the underlying diffusion models. Quantitative results can be found in Table 1 which further imply that the choice of sampler may be responsible for prior failures in compositional generation with diffusion models.

## 4.2 ENERGY-BASED PARAMETERIZATION

As noted in Section 3, we are unable to use mixture composition without an explicitly parameterized likelihood function. But, if we parameterize a potential function $f_\theta(x, t)$ and implicitly define $\epsilon_\theta(x, t) = \nabla_x f_\theta(x, t)$ we can recover an explicit estimate of the (unnormalized) log-likelihood – enabling us to utilize all presented forms for model composition.

Additionally, an explicit estimate of log-likelihood enables the use of more accurate samplers. As explained above, with the standard $\epsilon_\theta(x, t)$-parameterization we can only utilize unadjusted samplers. While they can perform well in practice, there exist many distributions from which they cannot generate decent samples (Roberts & Tweedie, 1996) such as targets with lighter-than-Gaussian tails where the ULA chain is transient. Additionally, for an accurate approximation to the Langevin SDE, ULA will need increasingly small stepsizes as the curvature of the log-likelihood gradient increases which can lead to arbitrarily slow mixing (Durmus & Moulines, 2019). In these settings a Metropolis correction can greatly improve sample quality and convergence. Again this issue can be solved by defining $\epsilon_\theta(x, t) = \nabla_x f_\theta(x, t)$ for some explicitly defined scalar potential function $f_\theta(x, t)$.

Energy-based parameterizations have been explored in the past (Salimans & Ho, 2021) and were found to perform comparably to score-based models for unconditional generative modeling. In that setting the score parameterization is then preferable as computing the gradient of the energy requires more computation. In the compositional setting, however, the additional flexibility enabled by explicit (unnormalized) log-probability estimation motivates a re-exploration of the energy-parameterization.

a) 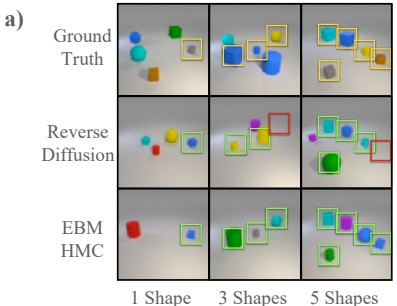

b) 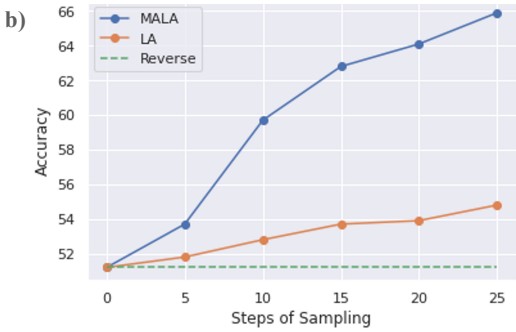

Figure 3: **(a) Composition enables the positions of multiple shapes to be simultaneously controlled, while training only conditions on the location of one object per image.** Reverse diffusion samples place shapes in incorrect locations. MCMC generates samples that satisfy all constraints. **(b) Metropolis adjustment significantly improves generation performance across sampling steps.** As more MCMC steps are run (at each timestep), generation accuracy of combinations of 5 cubes improves significantly.

We explored a number of energy-based parameterizations for diffusion models and ran a pilot study on ImageNet. In this study we found it best to parameterize the log probability as $f_\theta(x, t) = -||s_\theta(x, t)||^2$, where $s_\theta(x, t)$ is a vector-output neural network, like those used in $\epsilon_\theta(x, t)$-parameterized diffusion models. Full details on our study can be found in Appendix E. From here on, all energy-based diffusion models take the above form.

Our energy-parameterized models enable us to use MALA and HMC samplers which produce our best compositional generation results by a large margin. An additional benefit of these samplers is that, through monitoring their acceptance rates, we are able to derive an effective automated method for tuning their hyper-parameters (a notoriously difficult task prior) which is not available for unadjusted samplers. Details of our samplers and tuning procedures can be found in Appendix C.1.

## 5 EXPERIMENTS

We experiment with various model parameterizations and sampling schemes for compositional generation with diffusion models. We first investigate these ideas on some illustrative 2D datasets, then move to the image domain with an artificial dataset of shapes. Here, we compose a model conditioned on the location of a single shape with itself to condition on the location of all of the shapes in the image. After this we experiment with classifier guidance on the ImageNet dataset. Finally, we self-compose text-to-image models to generate from compositions of various text prompts. Full details of all experiments can be found in Appendix G. Throughout we compare our proposed improvements with a score-parameterized model using standard reverse diffusion sampling. We note that this baseline is exactly the approach of Liu et al. (2022).

### 5.1 2D DENSITIES

We train diffusion models using both parameterizations and study the impact of various sampling approaches for compositional generation. Samples are evaluated using RAISE (Burda et al., 2015) (which gives lower bounds on log-likelihood) and MMD[2], LL (log-likelihood of generated samples under composed distribution), and Var (L2 difference of variance of GMMs fit on generated samples compared to GMMs of the composed distribution). Results can be found in Table 1 and visualizations can be seen in Figure 2. All MCMC sampling methods improve sample quality and likelihood, with Metropolis adjusted methods performing the best. All MCMC experiments use the same number of score function evaluations. We include a baseline, labeled "Reverse (equal steps)" which is a diffusion model trained with more steps such that reverse diffusion sampling has the same cost as our MCMC samplers. We see that simply adding more time-steps does not solve compositional sampling.

### 5.2 COMPOSING CUBES

Next, we train models on a dataset of images containing between 1 and 5 examples of various shapes taken from CLEVR (Johnson et al., 2017). We train our models to fit $p(x|y)$ where $y$ is the location of *one* of the shapes in the image. We then compose this conditional model with itself to create a product model which defines the distribution of images conditioned on $c$ shapes as

---

[2]For the mixture we use MMD to replace RAISE likelihood based evaluation as we encountered numerical stability issues with RAISE when applying to the mixture.

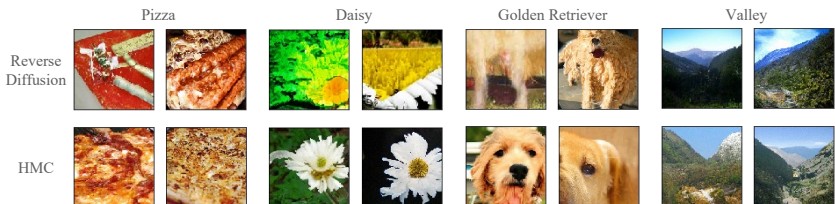

Figure 4: Classifier-guided generation on ImageNet. **HMC leads to higher fidelity and more class-identified images than reverse diffusion sampling.**

| Model | Sampler | Combinations | | | | |
|-------|---------|------|------|------|------|------|
| | | 1 | 2 | 3 | 4 | 5 |
| Score | Reverse | 70.8 | 68.2 | 66.3 | 64.1 | 57.4 |
| | ULA | 75.0 | 73.4 | 71.8 | 67.9 | 60.2 |
| | U-HMC | **79.1** | **76.0** | **73.6** | **71.1** | **62.3** |
| EBM | Reverse | 71.0 | 67.1 | 62.5 | 58.1 | 51.0 |
| | ULA | 81.3 | 71.8 | 66.6 | 59.6 | 54.8 |
| | MALA | 85.4 | 74.4 | 71.1 | 65.6 | 63.9 |
| | U-HMC | 84.5 | 81.3 | 79.2 | 74.2 | 68.1 |
| | HMC | **91.6** | **82.9** | **80.1** | **76.5** | **72.7** |

Table 2: **MCMC Sampling enables more compositional cube generation on CLEVR.**

| Model | Sampler | Inception Score ↑ | FID ↓ | Accuracy ↑ |
|-------|---------|-------------------|-------|------------|
| Score | Reverse | 29.10 | 30.46 | 18.64 |
| | LA | 29.35 | 30.49 | 65.81 |
| | U-HMC | **32.19** | **26.89** | **89.93** |
| EBM | Reverse | 28.05 | 33.58 | 18.60 |
| | LA | 28.12 | 33.45 | 66.28 |
| | MALA | 30.43 | 32.22 | 83.65 |
| | U-HMC | 31.39 | 32.08 | 90.83 |
| | HMC | **33.46** | **30.52** | **94.61** |

Table 3: **MCMC Sampling enables better classifier guidance on 128x128 ImageNet dataset.**

$$\log p_\theta(x|y_1, \ldots, y_c) = \log p_\theta(x) + \sum_{i=1}^{c} \left( \log p_\theta(x|y_i) - \log p_\theta(x) \right). \tag{13}$$

We then sample using various methods, where for each number of combination of cubes, the same number of score function evaluations are used, and evaluate each by the fraction of samples which have all objects placed in the correct location (as determined by a learned classifier). Results can be found in Table 2, where we see MCMC sampling leads to improvements and the Metropolis adjustment enabled by the energy-based parameterization leads to further improvements. We qualitatively illustrate results in Figure 3, and see more accurate generations with more steps of sampling, with more substantial increases with Metropolis adjustment.

### 5.3 CLASSIFIER CONDITIONING

Next, we train unconditional diffusion models and a noise-conditioned classifier on ImageNet. We compose these models as

$$\nabla_x \log p_\theta(x|y, t) = \nabla_x \log p_\theta(x|t) + \nabla_x \log p_\theta(y|x, t). \tag{14}$$

and sample using the corresponding score functions. We compare various samplers and model parameterizations on classifier accuracy, FID (Heusel et al., 2017) and Inception Score. Quantitative results can be seen in Table 3 and qualitative results seen in Figure 4. We find that MCMC improves performance over reverse sampling, with further improvements from Metropolis corrections.

### 5.4 TEXT-2-IMAGE

Perhaps the most well-known results achieved with diffusion models are in text-to-image generation (Ramesh et al., 2022; Saharia et al., 2022). Here we model $p_\theta(x_{\text{image}}|y_{\text{text}})$. While generated images generated are photo-realistic, they can fail to generate images from prompts which specify multiple concepts at a time (Liu et al., 2022) such as $y_{\text{text}} = $ "A horse on a sandy beach or a grass plain on a not sunny day". To deal with these issues we can dissect the prompt into smaller components $y_1, \ldots, y_c$, parameterize models conditioned on each component $p_\theta(x|y_i)$ and compose these models using our introduced operators. We can parse the above example into

"A horse" AND ("A sandy beach" OR "Grass plains") AND (NOT "Sunny")

which can be used to define the following (unnormalized) distribution

$$p_\theta^{\text{comp}}(x|y_{\text{text}}) \propto \frac{p_\theta(x|\text{"A horse"}) \left[ \frac{1}{2} p_\theta(x|\text{"A sandy beach"}) + \frac{1}{2} p_\theta(x|\text{"Grass plains"}) \right]}{p_\theta(x|\text{"Sunny"})^\alpha}$$

Liu et al. (2022) demonstrated that composing models this way can improve the efficacy of these kinds of generations, but was restricted to composition using classifier-free guidance. We train a

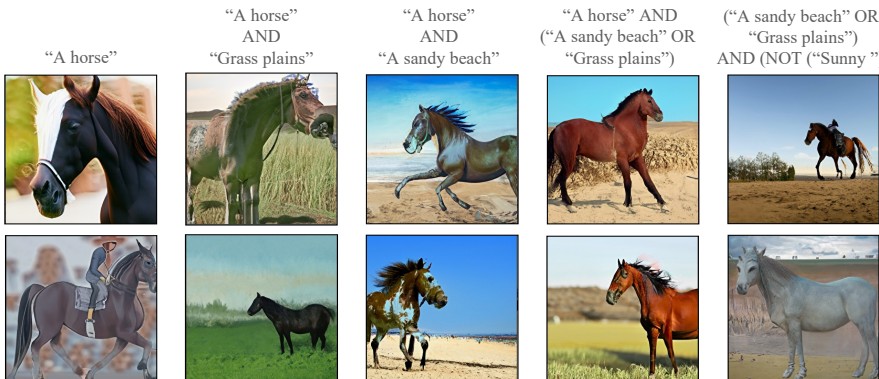

Figure 5: **Energy based parameterization enables high-resolution compositional text-to-image synthesis.**

energy-parameterized diffusion model for text conditional 64x64 image generation and illustrate composed results in Figure 5 (upsampled to 1024x1024). We find that composition enables more faithful generations of scenes in Figure 6 with more results in Appendix A.

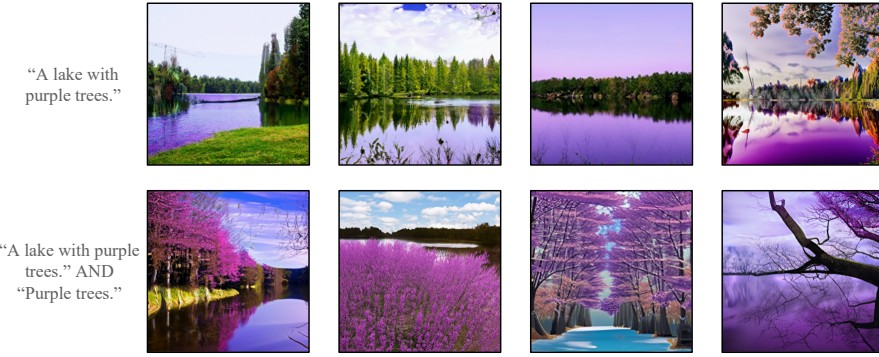

Figure 6: **Composing text descriptions enables more accurate scene generation.**

## 6    DISCUSSION

**Limitations.**    Our work demonstrates that diffusion models, in combination with MCMC-based sampling procedures, can be composed in novel ways capable of generating high-quality samples. However, our proposed solutions have a number of drawbacks. First, more sophisticated MCMC samplers come at a higher cost than the standard sampling approach and can take 5-times longer to generate samples than typical diffusion sampling. Second, we have shown that energy-parameterized models enable the use of more sophisticated sampling techniques, garnering further improvements. Unfortunately, this requires a second backward-pass through the model to compute the derivative implicitly, leading them to have double the memory and compute cost of score-parameterized models.

While these are considerable drawbacks, we note the focus of this work is to demonstrate that such things are possible within the framework of diffusion models. We believe there is much that can be done to achieve the benefits of our sampling procedures at less cost such as distillation (Salimans & Ho, 2022) and easier-to-differentiate neural networks (Chen & Duvenaud, 2019).

Finally we note that not all models can be effectively composed together. For example if we wanted to model the product of $\mathcal{N}(-10, 1)\mathcal{N}(10, 1)$, the resulting distribution's support is far outside the support of the constituent models. To accurately model this, our constituent models would need to be near-perfect far outside the training distribution. Thus it is unlikely for good results to be obtained. The same care should be taken in the text-2-image setting with, for example, contradicting prompts.

**Conclusion.**    In this work we have explored the ways that pretrained diffusion models can be composed to model new distributions. We demonstrate ways that naïve implementations fail, and present two ways that performance can be improved: MCMC sampling and energy-parameterized diffusion models. We find our proposed methods lead to notable improvement across a variety of domains, scales, and different compositional operators.

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

# Appendix

In this appendix, we present additional text-to-image results in Section A. We present detailed derivations of diffusion models in Section B. We present additional information on MCMC sampling in Section C.1. We provide additional derivations on composing diffusion models in Section D. We discuss different parameterizations of energy based diffusion models in Section E. We further provide additional example 2D compositions in Section F. Finally, we provide experimental details in Section G.

## A  TEXT-TO-IMAGE RESULTS

We present additional use cases of composing models in different text-to-image domains. First, in Figure A1, we illustrate how composing two separate energy parameterized diffusion models enables us to more accurately generate images that have more detailed information in the caption. Next, in Figure A2, we illustrate how composing two separate energy parameterized diffusion models further enable us to accurately generate images with the correct colors assigned to each object. We further show in Figure A3 how composing the negation of one energy parameterized diffusion model with another other enables us to generate images where one commonly occurring co-founding factor does occur (i.e. a sandy beach without coastal water). Finally, we illustrate in Figure A4, how composing multiple diffusion models enables us to render the number of objects in a scene accurately.

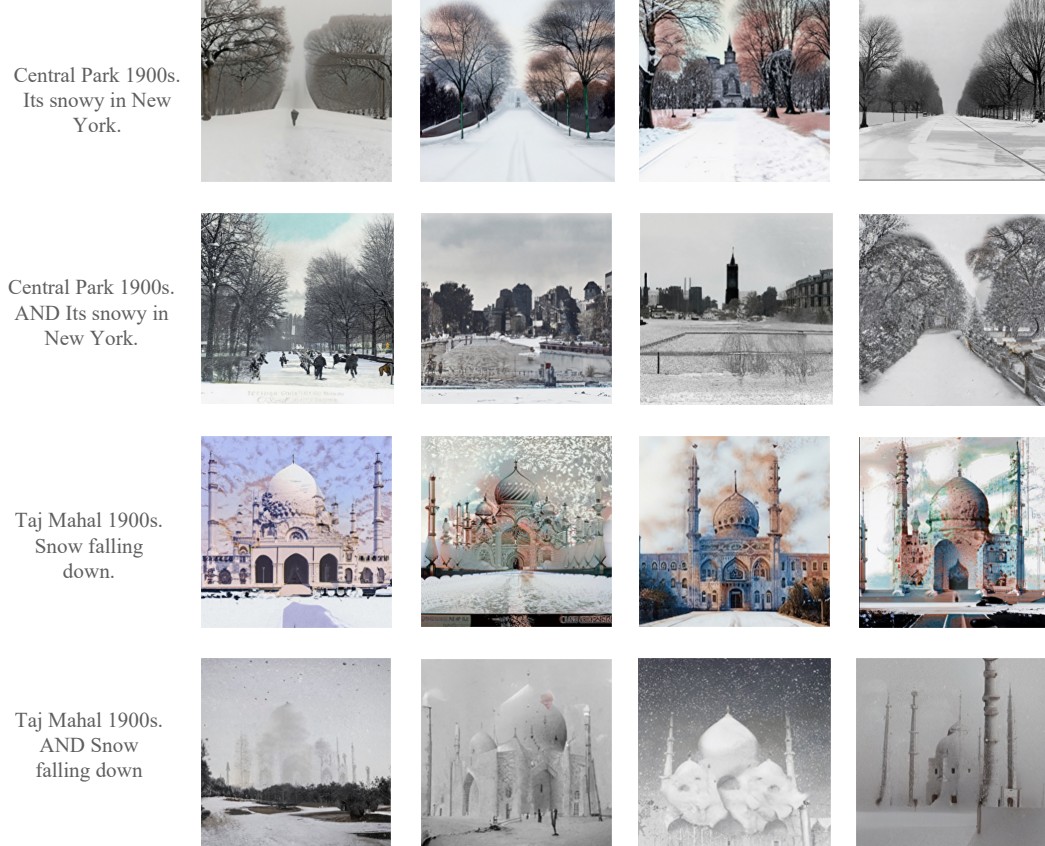

Figure A1: By composing energy based diffusion models, we can render more detailed information in images. In the above images, we can more accurately render details such as Central Park (top) or the effect of snowing (bottom).

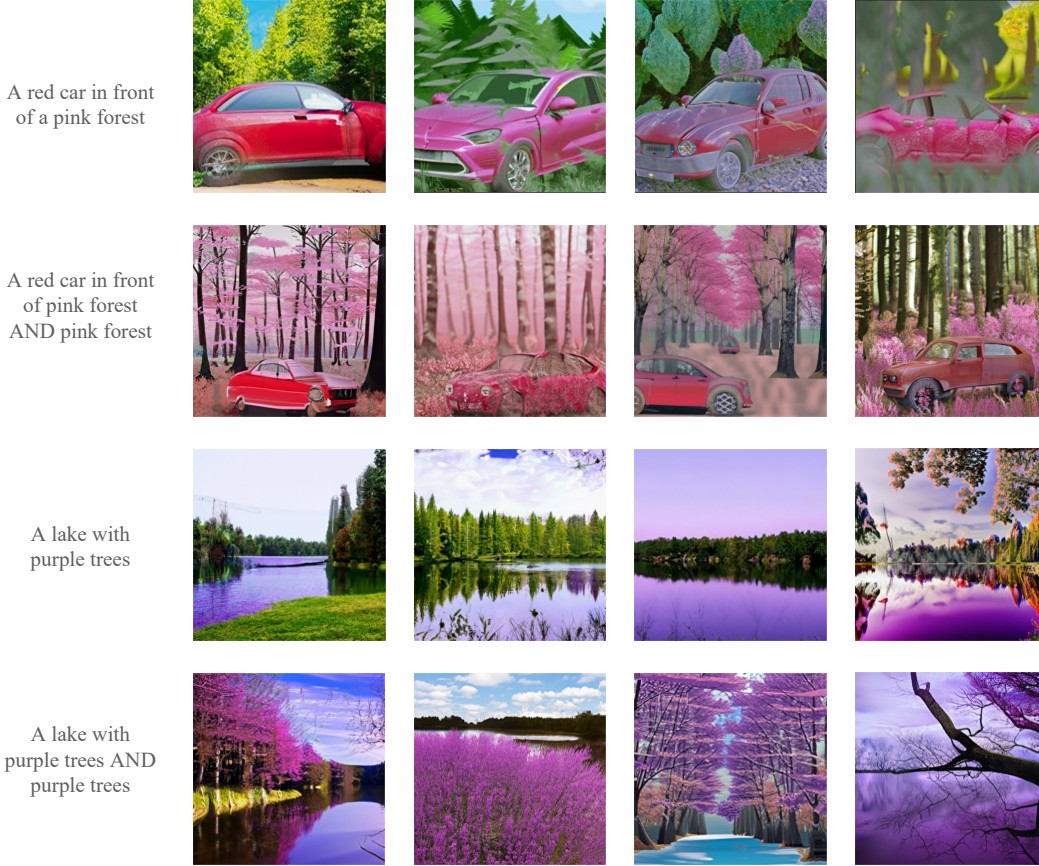

Figure A2: By composing energy based diffusion models, we can more accurately render different colors of objects in a scene.

## B  DETAILED DERIVATION OF DIFFUSION MODELS

Diffusion models seek to model a data distribution $q(x_0)$ (written this way for notational convenience) and define a series of latent variables $x_1, \ldots, x_T$ generated from a Markov process $x_t \sim q(x_t|x_{t-1})$ where

$$q(x_t|x_{t-1}) = \mathcal{N}\left(x_t; \sqrt{1 - \beta_t}\, x_{t-1}, \beta_t I\right). \tag{A1}$$

A unique and useful property of this process is that all time marginals $q(x_t|x_0)$ can be computed in closed form and are Gaussian

$$q(x_t|x_0) = \mathcal{N}\left(x_t; \sqrt{1 - \sigma_t^2}\, x_0, \sigma_t^2 I\right). \tag{A2}$$

where $\sigma_t^2 = 1 - \bar{\alpha}_t$ and $\bar{\alpha}_t = \prod_{t=1}^{T}(1 - \beta_t)$. We can see that if all $\beta_t > 0$, then as $t \to \infty$ $q(x_t|x_0) \to \mathcal{N}(x_t; 0, I)$.

We seek to train a model $p_\theta(x_{t-1}|x_t)$ which reverses $q(x_t|x_{t-1})$ step-wise with a parametric model. We can analytically derive the variance of the reversal as $\tilde{\beta}_t = \frac{1 - \bar{\alpha}_{t-1}}{1 - \bar{\alpha}_t}$ and define

$$p_\theta(x_t|x_{t+1}) = \mathcal{N}(x_t; \mu_\theta(x_{t-1}, t), \tilde{\beta}_t I) \tag{A3}$$

and set $p(x_T) = \mathcal{N}(0, I)$. We train this model to maximize a variational bound on the marginal likelihood

$$\log p_\theta(x_0) \geq \mathbf{E}_{q(x_1, \ldots, x_T|x_0)}[\log p_\theta(x_0|x_1) + \sum_{t=1}^{T} D_{KL}(q(x_t|x_{t-1}, x_0)||p(x_t||x_{t-1})) \tag{A4}$$

$$+ D_{KL}(q(x_T|x_0)||p(x_T))]. \tag{A5}$$

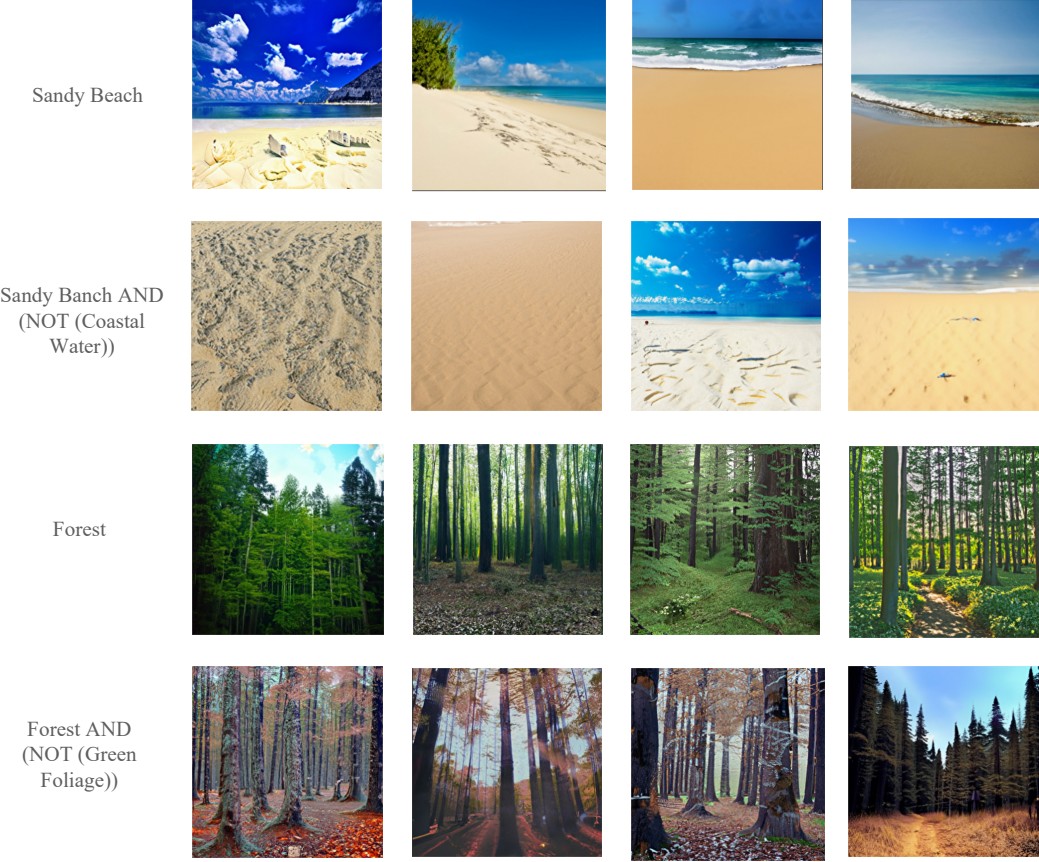

Figure A3: By composing an energy parameterized diffusion model with the negation of another energy parameterized diffusion model, we can render images in unusual configurations not typically found in the data.

The first term lacks parameters and the final is approximately 0 from the convergence of the $q$-process so we focus on the middle terms.

Typically, the model $\mu_\theta(x_{t-t}, t)$ is not parameterized to predict the mean of $x_t$. Instead it is parameterized to predict the noise added to $x_t$ to arrive at $x_{t-1}$. This motivates the following parameterization

$$\mu_\theta(x_t, t) = \frac{1}{\sqrt{\alpha_t}} \left( x_t - \frac{\beta_t}{\sqrt{1 - \bar{\alpha}_t}} \epsilon_\theta(x_t, t) \right). \tag{A6}$$

In this form we can rewrite the important terms in the objective as

$$D_{KL}(q(x_t|x_{t-1}, x_0)||p(x_t||x_{t-1})) = -C_t \mathbf{E}_{q(x_0)\mathcal{N}(\epsilon; 0, I)} \left[ ||\epsilon - \epsilon_\theta(x_t, t)||^2 \right] = C_t \mathcal{L}(x, \sigma) \tag{A7}$$

where $C_t$ is a time-dependent constant. Typically these are dropped and all objectives are weighted equally.

Once we finish training, we can draw samples from our model by first sampling $x_T \sim p(x_T)$ and then equentially sampling $x_t = \mu_\theta(x_{t-1}, t) + \sqrt{\tilde{\beta}_t}\epsilon$ where $\epsilon \sim \mathcal{N}(0, I)$.

## C   MCMC SAMPLING DETAILS

### C.1   HAMILTONIAN MONTE-CARLO AND ITS VARIANTS

Hamiltonian Monte-Carlo (Neal, 1996) seeks to sample from an unnormalized probability distribution $\log p(x) = f(x) + \log Z$. To do this, we augment our distribution over $x$ with auxillury variables $v$

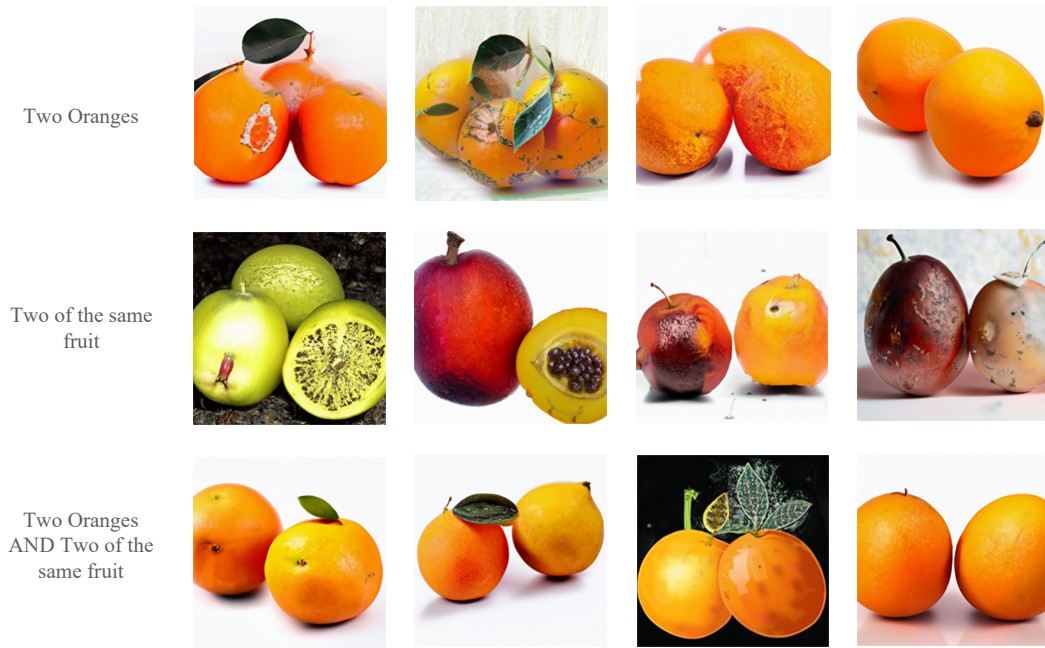

Figure A4: By composing multiple energy parameterized diffusion models, we can more accurately render the underlying number of objects in ascene.

and define the joint distribution $p(x, v) = p(x)\mathcal{N}(v; 0, M)$ where the covariance $M$ is known as the "mass-matrix." We now seek to draw samples $x, v \sim p(x, v)$ and since $x$ and $v$ are independent under the joint, we can simply throw away our $v$ samples leaving us with a sample $x \sim p(x)$.

Like other MCMC methods we sequentially update a particle $(x^i, v^i)$ in such a way that as $i \to \infty$ we arrive at a sample from $p(x, v)$. For a step of HMC, starting at $(x^i, v^i)$ we first sample $v^{i\prime} \sim \mathcal{N}(v^i; 0, M)$ since the target distribution factorizes and $p(v)$ is known and tractable. We then integrate a likelihood-conserving ODE defined on $x, v$ known as "Hamiltoninan Dynamics." We can use the likelihood-preserving leapfrog integrator which will guarentee that the transition distribution is symmetric, i.e $k(x', v'|x, v) = k(x, v|x', v')$. Thus, the Metropolis acceptance probability simplifies to $\min\left(1, \frac{p(x', v')}{p(x, v)}\right)$. An overview of the HMC algorithm can be found in Algorithm 2. We refer the reader to Neal (1996) for a more complete description of the algorithm.

---

**Algorithm 2** Hamiltonian Monte-Carlo

---

**Input:** Initial state $x^0$, Mass matrix $M$, Number of steps $N$, Number leapfrog steps $L$, step-size $\epsilon$
**for** $i = 1, \ldots, N$ **do**
    Sample $v^i \sim \mathcal{N}(0, M)$      # Sample momentum
    $x', v^{i\prime} = \text{Leapfrog}(x^{i-1}, v^i; \epsilon, L)$      # Integrate dynamics with stepsize $\epsilon$ for $L$ steps
    $a = \min\left(1, \frac{p(x', v^{i\prime})}{p(x^{i-1}, v^i)}\right)$      # Compute acceptance probability
    With probability $a$
        set $x^i = x'$
    else
        set $x^i = x^{i-1}$
**end for**
**return** $x^N$

---

Since our $\epsilon_\theta(x, t)$ parameterized models do not admit an explicit likelihood function, we use an unadjusted variant of HMC (U-HMC) where the accept/reject step is simply ignored.

We can see in Algorithm 2 that at every step, the momentum is re-sampled. This can be sub-optimal, as the momentum determines the initial direction of $x$'s movement and if a good direction is found, it may be beneficial to continue in that direction. To deal with this, Neal (1996) proposes a variant of HMC where the momentum $v$ is partially retained between sampling steps. We add an additional sampler parameter $\gamma \in [0,1]$ (known as the "damping-factor") which controls the amount to which $v$ is retained. When $\gamma$ is close to 1, $v$ is mostly kept and when it is near 0, $v$ is mostly refreshed. This variant is summarized in Algorithm 3. The potentially confusing momentum negations ensure the validity of the sampler. Intuitively, when the proposal is accepted, the momentum is retained and when it is rejected the momentum is flipped. For this reason, one should maintain a reasonably high acceptance rate when using this approch.

---

**Algorithm 3** Hamiltonian Monte-Carlo with Partial Momentum Refreshment

---

**Input:** Initial state $x^0$, Mass matrix $M$, Number of steps $N$, Number leapfrog steps $L$, step-size $\epsilon$, damping-factor $\gamma$

Sample $v^0 \sim \mathcal{N}(0, M)$     # Sample initial momentum
**for** $i = 1, \ldots, N$ **do**
   $\lambda \sim \mathcal{N}(0, M)$
   $v^{(i-1)\prime} = \gamma v^{i-1} + \sqrt{1 - \gamma^2}\lambda$     # Partially refresh momentum
   $x', v' = \text{Leapfrog}(x^{i-1}, v^{(i-1)\prime}; \epsilon, L)$     # Integrate dynamics with stepsize $\epsilon$ for $L$ steps
   $v' = -v'$    # Negate momentum
   $a = \min\left(1, \frac{p(x', v')}{p(x^{i-1}, v^{(i-1)\prime})}\right)$     # Compute acceptance probability
   With probability $a$
       set $x^i = x'$, $v^i = v'$
   else
       set $x^i = x^{i-1}$, $v^i = v^{(i-1)\prime}$
   $v^i = -v^i$    # Negate momentum
**end for**
**return** $x^N$

---

## C.2 MCMC Tuning

A crucial component to ensure successful MCMC sampling in diffusion models is the choice of step sizes for samplers. We initialize step sizes for all samplers at each distribution $t$ to be roughly proportional to the $\beta_t$ noise values added to distribution $t$ in the diffusion process.

To tune step sizes across timesteps for both HMC and MALA samplers, to set step sizes at each timestep $t$ to be constant multiplied by $\beta_t$. We searched different constants to multiply $\beta_t$, and chose a value so that the average acceptance rate of MALA and HMC samplers across timesteps is approximately 60% and 70% respectively. For un-adjusted variants of these samplers, we set step sizes to be the same as adjusted samplers, and found limited gains when step sizes were specifically tuned towards the un-adjusted samplers. We utilize a mass matrix of $\beta_t$ for HMC samplers.

Precise details on the exact MCMC steps sizes used in experiments can be detailed in Section G.

## C.3 MCMC Implementation Details

When initially running MCMC sampling on diffusion models in the image domain, we found that our samplers tended to converge to images which had uniform textures. After experimentation, we found that the primary cause of this issue was fact that by default, typical implementations of the reverse diffusion process clip samples at intermediate time-steps of sampling to be between -1 and 1. To enable proper MCMC sampling, we found that it was important to *not clip* intermediate values of diffusion sampling.

When running MCMC sampling on image domains, we further found that it was helpful for mixing to run a single step of the reverse process to initialize MCMC sampling, before running many steps of MCMC sampling at each timestep $t$, and in all MCMC sampling settings on the image domain, we run one step of the reverse process before running MCMC sampling. Such a MCMC sampling

procedure is similar to the predictor-corrector sampling procedure introduced in (Song et al., 2021) for alleviating discretization errors when sampling continuous time diffusion models.

# D  COMPOSITIONAL DIFFUSIONS

In Equations 11 and 12, we demonstrate that for diffused distributions $\{q_t^i(x_t)\}$ where $q_t^i(x_t) = \int q^i(x_0)q(x_t|x_0)dx_0$, the diffusion of the product of $q^i$'s is not the same as the product of the diffusions, meaning plugging the product of diffusions into standard reverse diffusion sampling will not draw samples from the product model. We present similar results for tempering and predictive model composition.

## D.1  SAMPLING FROM A TEMPERED VERSION OF $q$ USING DIFFUSION?

It is tempting to believe that we can sample from a tempered/annealed version of the data distribution

$$q^\lambda(x) \propto q(x)^\lambda$$

using the tempered diffused data distribution $\lambda\nabla\log_t q(x_t)$ but this is incorrect. For this procedure to be correct, we would need to have $\nabla\log q_t^\lambda(x_t) = \lambda\nabla\log q_t(x_t)$ for all $t$. However, while we do have $\nabla\log q_0^\lambda(x_0) = \lambda\nabla\log q_0(x_0)$, this equality does not hold for $t > 0$

$$
\begin{aligned}
\nabla\log q_t^\lambda(x_t) &= \nabla\log\int q^\lambda(x_0)q(x_t|x_0)dx_0 \\
&\neq \lambda\nabla\log\int q(x_0)q(x_t|x_0)dx_0 \\
&= \lambda\nabla\log q_t(x_t).
\end{aligned}
$$

## D.2  GUIDANCE

For conditional generation, we should use in the reverse diffusion the score of the diffused conditional distribution $\nabla\log q_t(x_t|y)$ where

$$q_t(x_t|y) = \int q(x_0|y)q(x_t|x_0)dx_0.$$

We also have

$$\nabla\log q_t(x_t|y) = \nabla\log q_t(x_t) + \nabla\log q_t(y|x_t)$$

so that

$$\nabla\log q_t(y|x_t) := \nabla\log q_t(x_t|y) - \nabla\log q_t(x_t)$$

allows you to do guidance without having to train say a classifier if $y$ is categorical.

In practice, it was found that using in the reverse time diffusion the score

$$\nabla\log q_t(x_t) + \lambda\nabla\log q_t(y|x_t)$$

generates much nicer images for $\lambda > 1$. However, it is also often claimed that it samples from a modified posterior where the likelihood has been annealed. This is incorrect. For a modified posterior with annealed likelihood, we would have

$$q^\lambda(x_0|y) \propto q(x_0|y)\{q(y|x_0)\}^\lambda$$

and it is not true again that

$$
\begin{aligned}
\nabla\log q_t^\lambda(x_t|y) &= \nabla\log\int q^\lambda(x_0|y)q(x_t|x_0)dx_0 \\
&\neq \nabla\log q_t(x_t) + \lambda\nabla\log q_t(y|x_t).
\end{aligned}
$$

### D.3    SAMPLING FROM COMPOSED DISTRIBUTIONS

We can see that products, tempering, and guidance applied to diffused distributions do not give diffusions of the modified target distributions. Thus, we should not expect to arrive at our desired result by applying a sampling procedure which reverses a diffusion applied to the target distribution. Thankfully, as stated in Section 4, these operators do give us a sequence of distributions which anneals from $\mathcal{N}(0, I)$ to the composed target which means we can utilize the family of annealed MCMC sampling methods mentioned in Section 4.1 to draw samples from our composed models in all of these settings, directly using the available score estimate.

## E    ENERGY-BASED PARAMETERIZATIONS

As mentioned in section 4.2, when using the $\epsilon_\theta(x, t)$ parameterization, we can recover an estimate of the time-conditional score function with $\nabla_x \log p_t(x) \approx -\frac{\epsilon_\theta(x,t)}{\sigma_t}$. This estimate of the log-likelihood gradient can be used for MCMC sampling methods which only require the log-likelihood gradient – such as ULA or U-HMC. These methods can work well, but will never generate exact samples when using non-zero step-sizes. Exact samplers can be derived from approximate samplers like the above methods using Metropolis corrections. Unfortunately, even if the samplers' transition distribution $k_i(\cdot, \cdot)$ does not require $\log p_\theta(x_t)$ evaluation, the Metropolis correction probability:

$$\min\left(1, \frac{e^{f_\theta(\hat{x})}}{e^{f_\theta(x_{t-1})}} \frac{k(x_{t-1}|\hat{x})}{k(\hat{x}|x_{t-1})}\right)$$

does. Futhermore, when we only have an estimate of the score at our disposal, we are only able to compose models using products.

To enable the use of Metropolis corrections and more compositional operators, we propose to change the parameterization of our diffusion model. Instead of using a neural net $\epsilon_\theta(x, t) : \{\mathbb{R}^d \times \mathbb{N}\} \to \mathbb{R}^d$, we define a scalar-output neural network $f_\theta(x, t) : \{\mathbb{R}^d \times \mathbb{N}\} \to \mathbb{R}$. We then compute the gradient of this function and define $\epsilon_\theta(x, t) = \nabla_x f_\theta(x, t)$. From here, we use this implicitly-defined $\epsilon_\theta(x)$ as in standard diffusion model training. As before, we can recover $\nabla_x \log p_t(x) \approx -\frac{\epsilon_\theta(x,t)}{\sigma_t}$, but now we are also able to recover $\log p_t(x) \approx -\frac{f_\theta(x,t)}{\sigma_t} + \log Z$ which enables the application of Metropolis corrected sampling.

Much prior work on EBMs parameterizes $f_\theta(x, t)$ using a feed-forward neural network, whose final layer has a single output (Nijkamp et al., 2020; Du & Mordatch, 2019). Salimans & Ho (2021) compare this approach with the standard $\epsilon_\theta(x, t)$ parameterization and find the $\epsilon_\theta(x, t)$ parameterization to perform better for unconditional image generation. We believe this has to do with the relative sparsity of the gradients of feed-forward neural networks. This can cause difficulties when training to optimize a function of their implicitly computed gradients.

Intriguingly, Salimans & Ho (2021) also explore a more structured energy function definition inspired by denoising autoencoders:

$$f_\theta^{DAE}(x, t) = -\frac{1}{2}||x - s_\theta(x, t)||^2$$

where $s_\theta(x, t) : \{\mathbb{R}^d \times \mathbb{N}\} \to \mathbb{R}^d$ is a neural network (identical to the standard $\epsilon_\theta(x, t)$) model. We can simply evaluate the gradients of this function to obtain

$$\nabla_x f_\theta^{DAE}(x, t) = (x - s_\theta(x, t)) - (x - s_\theta(x, t))\nabla_x s_\theta(x, t).$$

In their study, this parameterization was found to perform near identically to the $\epsilon_\theta(x, t)$ parameterization while admitting an explicit energy function. We believe this energy parameterization performs better because its gradients include the feed-forward network $s_\theta(x, t)$, making optimization easier. Salimans & Ho (2021) conclude that the $\epsilon_\theta(x, t)$ parameterization should be favored since the computing $\nabla_x f_\theta^{DAE}(x, t)$ requires computing $\nabla s_\theta(x, t)$ which requires an extra backward pass through the neural network, increasing compute.

We reexamine this energy parameterization and two other choices now that our application motivates having access to an explicit energy function. The other parameterizations are based different transformations of the $s_\theta(x, t)$ architecture; the negative L2 norm (L2) and an inner product (IP). They are

defined as:

$$
\begin{aligned}
f_\theta^{L2}(x,t) &= -\frac{1}{2}||s_\theta(x,t)||^2 \\
\nabla_x f^{L2}(x,t) &= -s_\theta(x,t)\nabla_x s_\theta(x,t)
\end{aligned}
$$

and:

$$
\begin{aligned}
f_\theta^{IP}(x,t) &= x^T s_\theta(x,t) \\
\nabla_x f^{IP}(x,t) &= s_\theta(x,t) + x^T \nabla_x s_\theta(x,t).
\end{aligned}
$$

We train models with each parameterization on ImageNet and compare using FID for unconditional sampling. Results can be seen in Table A1. We see that L2 and Inner-Product perform the best, but are both outperformed by the standard parameterization. We initially experimented with these two parameterizations but found that the L2-norm parameterization to be more stable for compositional sampling due, we believe, to the fact that the energy-function is bounded above meaning that MCMC sampling is incapable of running off to infinity to increase likelihood.

| Parameterization | | | |
|:---:|:---:|:---:|:---:|
| (1) DAE | (2) L2 Norm | (3) Inner-Product | $\epsilon_\theta(x,t)$ |
| 97.4 | 91.5 | **90.9** | 86.7 |

Table A1: FID (1k samples) of various energy-parameterizations on unconditional ImageNet generation.

## F  SYNTHETIC DISTRIBUTION COMPOSITIONS

**Mixture**   We provide additional 2D illustrations of mixtures of two diffusion models in Figure A5. We find that HMC sampling enables more accurate mixtures of different synthetic distributions.

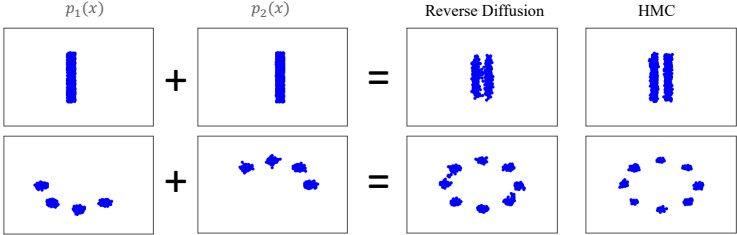

Figure A5: **Examples of mixture applied to diffusion models.** Left to right: Component distributions, reverse diffusion, HMC sampling. Reverse diffusion fails to sample accurately from mixed distributions distributions.

**Negation**   We provide additional 2D illustrations of negating two diffusion model with respect to each other in Figure A7. We find that HMC sampling enables accurate negations of different sythetic distributions.

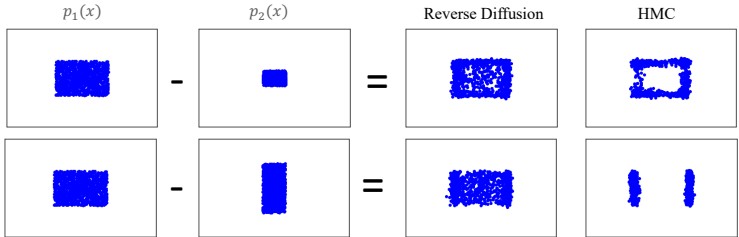

Figure A6: **Examples of negation applied to diffusion models.** Left to right: Component distributions, reverse diffusion, HMC sampling. Reverse diffusion fails to sample accurately from negated distributions.

**Failure Cases**   Next we illustrate a failure case of composition using our approach in Figure A7. Our approach fails to generate the product of two distribution when they are disjoint with respect to each other.

Figure A7: **Failure Cases of Our Approach.** Left to right: Component distributions, reverse diffusion, HMC sampling. Our approach fails to generate products of distributions with no overlap.

## G  EXPERIMENTAL DETAILS

We provide detailed experimental details including underlying quantitative metrics, training details, and architectures on 2D synthetic, CLEVR, ImageNet, and test-to-image settings below. To enable stable training of energy-based diffusion models in image settings, we clip gradient norms to be less than 10, and initialize convolutional layers using zero-initialization (Zhang et al., 2019).

**Synthetic Datasets**    For synthetic datasets, we train both score and energy based diffusion models using a small residual MLP model with 4 residual blocks, with a internal hidden dimension of 128 dimensions. We train models for 15000 iterations (10 minutes on a 8 TPUv2 cores) using the Adam optimizer with learning rate of 1e-3, and train diffusion models on 100 discrete timesteps with linear schedule of $\beta$ values.

When evaluating product of diffusion models, we generate two separate distributions, where train two separate diffusion models. In our first distribution, we construct a GMM of 8 Gaussians in a ring of radius 0.5 around the origin, with each Gaussian having a standard deviation of 0.3. In our second dataset, we construct a uniform distribution of points with $x$ between -0.1 and 0.1 and $y$ between -1 and 1. When evaluating mixture diffusion models, we generate one distribution consisting of a mixture of 3 Gaussian with standard deviation 0.03 and centers at $(-0.25, 0.5), (-0.25, 0.0), (-0.25, -0.5)$, and another distribution consisting of a mixture of 3 Gaussian with standard deviation 0.03 and centers at $(0.25, 0.5), (0.25, 0.0), (0.25, -0.5)$.

To construct MCMC samplers from models on synthetic datasets, we run 3 steps of HMC per timestep, with 3 leapfrog steps per step of HMC. We run 10 steps of MALA sampling per timestep. We found that MCMC performed robustly in the 2D dimensional setting and set the step size of MALA to be 0.002 across all distributions and the step size of HMC to be 0.03 across all distributions (with a mass matrix of 1)

**CLEVR**    For CLEVR, we generated a dataset of 200,000 $64 \times 64$ images with between 1 to 5 different cubes using dataset generation code in (Liu et al., 2021). To evaluate the accuracy in which generated images had cubes at each specified position, we trained a binary classifier on these images, and marked a cube as correctly generated if the confidence of the binary confidence of classifier is greater than 0.5.

To parameterize our diffusion architecture, we follow the architecture of (Ho et al., 2020), where we use a base hidden dimension of 128, and multiply the hidden dimensions by $[1, 2, 3, 4]$ at different resolutions of the image. We utilize 3 residual blocks at each resolution of the image. We trained diffusion models with 100 discrete timesteps with a linear $\beta$ schedule. CLEVR models were trained for 20000 iterations with a batch size of 1024 using the Adam optimizer with step size 1e-4, corresponding to roughly 8 hours on 8 TPUv2 cores.

To initialize MCMC sampling on the CLEVR domain, at each timestep, before applying MCMC sampling, we run one step of the reverse process in the trained diffusion model. We run 40 steps of MCMC sampling per timestep for MALA samplers, and 13 steps of HMC sampling (with 3 leapfrog step per HMC step) (with the mass matrix of HMC samplers set to $\beta$). We use HMC with partial momentum refreshment, and use a dampening coefficient of 0.9 across HMC iterations. MALA step sizes are set to $0.035 * \beta_t$, and HMC step sizes are set to $0.1 * \beta_t$

**ImageNet**    For ImageNet, we train an unconditional diffusion model $128 \times 128$ images. We train diffusion models for 1 million iterations of ImageNet with a batch size of 64 (3 days on 16 TPUv2

cores), using Adam optimizer with learning rate 1e-4, for 1 million iterations. We train diffusion models with 1000 discrete timesteps using the

On the ImageNet dataset, we report three seperate metrics. To report classifier accuracy, we feed generated sample into a ImageNet classifier trained on clean images, and label a image as correctly generated if the classifier of a generated image having the specified class is greater than 50%. We further report the Inception Score and FID, which are calculated on 50000 generated samples.

We follow the architecture of (Ho et al., 2020), where we use a base hidden dimension of 128 and multiply the hidden dimensions by $[1, 1, 2, 3, 4]$ at the different resolution of the image. We utilize 2 residual blocks at each resolution of the image.

To initialize MCMC sampling on the ImageNet domain, at each timestep, before applying MCMC sampling, we run one step of the reverse process in the trained diffusion model. We run 6 steps of MCMC sampling per timestep for MALA samplers, and 2 steps of HMC sampling (with 3 leapfrog steps per HMC step and with the mass matrix of HMC samplers set to $\beta$). MALA step sizes are set to $0.5 * \beta_t$ , and HMC step sizes are set to $0.6 * \beta_t^{1.5}$

**Text-to-Image** For text-to-image models, we train models for one week on an internal text/image dataset consisting of 400 million images using 32 TPUv3 cores, with a training data batch size of 256. We train our energy-based text-to-image model using a total of 1000 timesteps with a cosine beta schedule. We follow the architecture of (Ho et al., 2020), where we use a base hidden dimension of 256, and multiply the hidden dimensions by $[1, 2, 3, 4]$ at different resolution of the image. We utilize 3 residual blocks at each resolution of the image.

To upsample images from $64 \times 64$ resolution to $1024 \times 1024$ resolution, we utilize two trained unconditional diffusion models, one trained to upsample from $64 \times 64$ resolution to $256 \times 256$ resolution and one trained to upsample from $256 \times 256$ resolution to $1024 \times 1024$ resolution.

To initialize MCMC sampling on the text-to-image domain, at each timestep, before applying MCMC sampling, we run one step of the reverse process in the trained diffusion model. We ran 2 steps of HMC sampling per timestep, with 3 leapfrog step per HMC step and a mass matrix of HMC samplers set to $\beta$). We use HMC with partial momentum refreshment, and use a dampening coefficient of 0.9 across HMC iterations. HMC step sizes are set to $0.1 * \beta_t$

