# OpenReview forum: "Reduce, Reuse, Recycle: Compositional Generation with Energy-Based Diffusion Models and MCMC"
_ICLR.cc/2023/Conference — Submitted to ICLR 2023_

### Official Review · Reviewer_ndtR · 2022-10-24

**Confidence:** 3
**Correctness:** 3
**Technical Novelty And Significance:** 2
**Empirical Novelty And Significance:** 3
**Recommendation:** 5

**Clarity, Quality, Novelty And Reproducibility:**

- The paper is easy to follow, but the organization can be improved.
- The novelty of the paper seems limited.
- The code is not provided.

**Strength And Weaknesses:**

Strengths:
1. This paper has a good motivation. For diffusion models, compositionality can be an important property for many applications. For example, in the experimental results, the proposed method has shown good performance in the compositional scene generation and text-to-image generation.
2. This paper clearly explains the problems of diffusion models in compositional generation, and uses MCMC samplers which may incorporate Metropolis corrections by energy-based parameterized diffusion models. The effectiveness of the sampling procedure is validated in the experiments.

Weaknesses:
1. The novelty of the paper seems limited. All the building blocks of the proposed method are referred to existing works, including the composition operations in Section 3, MCMC samplers in Section 4.1 and energy-based parameterization in 4.2.
2. The organization of the paper can be improved. Preliminary knowledge (including introduction to diffusion models in Section 2 and introduction to composition operations in Section 3) span more than three pages. In contrast, the main technical part is too short to elaborate the method.
3. Although the paper points out the deviation between sampling from the composed score function and sampling from the composed diffusion model; however, this gap is not demonstrated on all tasks/datasets in the experiments.
4. In the abstract, it says “explore alternative ways to condition, modify, and reuse diffusion models” and “we investigate why certain types of composition fail using current techniques and present a number of solutions”; however it seems that there are no corresponding content in the main body of the paper, do I miss something or is it overstated in the abstract?

Some comments:
- Figure 3 lacks some necessary explanations, for example, what do the boxes with different colors indicate?
- In Table 2 and Table 3, are EBM and Energy the same model?
- Some good experimental results of text-to-image are shown in Figure 5. Some failure cases of this task can also be shown and discussed.

**Summary Of The Paper:**

This paper aims to compose energy-based diffusion models for compositional generation. The authors first point out the deviation between sampling from the composed score function and sampling from the composed diffusion model. To this end, MCMC samplers are adopted in the sampling procedure to improve the composition generation of diffusion models, and with proper energy-based parameterizations, Metropolis corrections may be incorporated into the sampling process. The experimental results demonstrate that the proposed method realizes the composition of diffusion models in different ways and on different data.

**Summary Of The Review:**

This paper is well motivated. It points out the deviation between sampling from the composed score function and sampling from the composed diffusion model. The novelty of the paper seems limited as all the building blocks of the proposed method are referred to existing works. The experimental results seem good.

---

> ### Author Response · Authors · 2022-11-11
> **Thank you for your constructive feedback (1/2)**
>
> We thank the reviewer for their time and feedback. We address your concerns in order.
>
> **Q1) The novelty of the paper**
>
> Please see our general response for a discussion of novelty. We would like to clarify our contributions of proposing MCMC for sampling from diffusion models and the novelty of the energy-based parameterization. While we agree that prior works on diffusion models have used MCMC and energy-based parameterizations, all prior works doing so do it in different ways with different final conclusions:
>
> - While [1] explores the energy-based parameterization, it concludes that there is no benefit to it. We show concretely that there are multiple benefits to using the energy-based parameterization when compositional generation is the goal. We clearly observe better results on all experiments.  Further, the energy-based parameterization enables the use of better MCMC samplers which was not mentioned in [1] (nor any prior work to our knowledge).
> - Regarding MCMC, the connections between diffusion models and MCMC sampling have long been known – in particular between standard reverse diffusion sampling and unadjusted Langevin dynamics. In most (if not all) prior work, it was assumed that unadjusted Langevin was a sufficient sampler for this class of model. In this work, we characterize the failure of compositional generation and hypothesize that this is a failure in sampling, not the model, motivating us to propose a new MCMC samplers that may be applied to diffusion models (Metropolis Adjusted / HMC samplers).
> - Our proposals for MCMC sampling is further reinforced by an investigation of the properties of reverse diffusion sampling from composed diffusion models (section 4 and Appendix D). We believe this to be a novel observation which motivates the application of more sophisticated samplers for compositional generation. We further provide sufficient evidence that this hypothesis is correct in our experiments which show across a variety of models and compositional operators that more advanced samplers lead to more successful compositional generation.
> - Finally, our approach further motivates the revisiting of energy-based parameterization for diffusion models (which the general community has lost interest in). Lastly, we present a novel form of energy-based model parameterization.
>
> **Q2) Organization**
>
> Thank you for this feedback. We have added additional detail on our method. Feel free to let us know additional things to add or which are not clear.
>
> **Q3) Empirical performance**
>
> We respectfully disagree with this comment. In all of our experimental results we demonstrate that improved sampling and energy-based parameterization improve compositional generation over reverse diffusion sampling. We present quantitative results on 1) toy data, 2) CLEVR shapes, 3) classifier conditioning on ImageNet. On each of these (tables 1, 2, 3, respectively), we observe considerable improvements from our proposed methods. Further, in our qualitative experiments we demonstrate high quality results on a compositional operator (mixtures) which can only be applied in the energy-based parameterization.
>
> **Q4) Abstract claims**
>
> Those topics you mention in the abstract are indeed in the paper. The first topic you mention is stated as:
>
> “This interpretation has motivated classifier-based and classifier-free guidance as methods for post-hoc control of diffusion models. In this work, we build upon these ideas using the score-based interpretation of diffusion models, and explore alternative ways to condition, modify, and reuse diffusion models for tasks involving compositional generation and guidance.”
>
> In Section 3 we explain that classifier-based and classifier-free guidance are a specific case of product composition. Next in this section, we explore alternative forms of composition, namely generalized products, mixtures, and negations. Of those, mixtures have not yet been demonstrated to be possible in the context of diffusion models.
>
> The next topic is introduced in the abstract as:
>
> “In particular, we investigate why certain types of composition fail using current techniques and present a number of solutions.”
>
> This is referenced in the main text in section 4, particularly in equations 11 and 12 (and Appendix D). There we show that composing diffused score functions does not necessarily give us the diffusion of the composed density. Thus, applying a sampling procedure which assumes a sequence of distributions corresponding to a gaussian diffusion of a given target should not necessarily give us the samples we want. We then demonstrate this in a number of quantitative and qualitative experiments.
>
> We believe both of these topics are addressed in the main body of the paper.
>
> We thank the reviewer for their feedback. We believe the new version of the paper is  notably improved.
>
> [1] Ho and Salimans, "Should EBMs model the energy or the score?", ICLR 2021 EBM Workshop

---

> > ### Author Response · Authors · 2022-11-11
> > **Thanks for the feedback (2/2)**
> >
> >
> > **Addressing your minor comments:**
> >
> > The boxes represent where the objects should be placed based on the conditioning. The boxes are colored green to indicate that the object is placed in the right location and red to indicate when the object is placed incorrectly. The top row are the ground truth. We gave them yellow boxes so as not to confuse them with the generated data in the bottom two rows. We will expand this caption to explain better in the next version of the paper.
> >
> > Yes EBM and Energy are the same. We have corrected Table 3 to say EBM instead. Thanks.
> >
> > We will add failure cases to the appendix and add a discussion of when things do and do not work. We are currently running these experiments and will follow up when they have been added to the appendix. We have added to the limitations section to explain situations where the methods proposed here may not work. Thank you.

---

> > > ### Author Response · Authors · 2022-11-16
> > > **Added new section**
> > >
> > > Hello!
> > >
> > > We would like to inform you that we have added a new section to the Appendix, section E which presents more 2D composition results and demonstrates failure cases as discussed in our previously added section.

---

> > ### Comment · Reviewer_ndtR · 2022-12-03
> > **Response**
> >
> > Thanks for the detailed response and further clarification on the contributions. After reading the revised paper, I'm still confused about comparison of the sampling results based on Eq.(11) and Eq.(12):
> >
> > - In Tables 1, 2, and 3, which results are obtained based on Eq.(11) and Eq.(12), respectively? Is that, except "reverse" based on Eq.(12), all the MCMC-based results are based on Eq.(11)? If so, the detail about how to sample based on Eq.(11) in the context of composing diffusion models is absent (not a general introduction to MCMC in Apendix C).
> >
> > - The performance gap between the samples based on Eq.(11) and Eq.(12) seems not been demonstrated for the text-2-image task (Section 5.3). In Figure 4, the composition operation is only conditioning on one class label, not the product of two or more conditions.

---

> > > ### Author Response · Authors · 2022-12-03
> > > **Clarifications**
> > >
> > > Hi Reviewer ndtR, thank you for taking time to read and respond to our response. Please see our clarifications below:
> > >
> > > > In Tables 1, 2, and 3, which results are obtained based on Eq.(11) and Eq.(12), respectively? Is that, except "reverse" based on Eq.(12), all the MCMC-based results are based on Eq.(11)? If so, the detail about how to sample based on Eq.(11) in the context of composing diffusion models is absent (not a general introduction to MCMC in Appendix C).
> > >
> > > Hi, we would like to clarify that Eq. 11 is the ideal reverse process we would like to sample from -- this is not possible however because we do not have the functional form for the expression inside it. Instead, Eq. 12 is a common approximation for Eq. 11 which is not exactly correct but corresponds to our reverse diffusion baseline. In our paper, we then propose MCMC based samplers which serves an accurate alternative for the reverse sampling procedure in Eq. 11.
> > >
> > > > The performance gap between the samples based on Eq.(11) and Eq.(12) seems not been demonstrated for the text-2-image task (Section 5.3). In Figure 4, the composition operation is only conditioning on one class label, not the product of two or more conditions.
> > >
> > > Our aim in results in the text-2-image results is to illustrate that our combined approach can scale to complex high-resolution text-2-image generative settings and thus we illustrate the qualitative results which are possible using our best approach. We are happy to add a figure comparing reverse diffusion and HMC sampling at this setting such as the one here https://ibb.co/7RBjgMs in the final version of the paper to further illustrate difference between using reverse diffusion and our MCMC sampler.
> > >
> > > The aim of our experiments in Figure 4 is to show that our approach can compose diffusion models with classifiers. We use the standard setting of classifier-based Imagenet generation where a single classifier is combined with a diffusion model. In this setting, it does not make as much sense to combine multiple conditions / classifiers as then we would be trying to generate an image which is both simultaneously "a daisy" and as well as "a dog".

---

> > > > ### Comment · Reviewer_ndtR · 2022-12-04
> > > > **Thanks**
> > > >
> > > > Dear authors,
> > > >
> > > > Thanks again for your further clarifications. I was afraid that I had missed some parts (e.g., being able to sample based on Eq.(11), comparison of the multi-condition composition results between the reverse sampling and the proposed MCMC methods in all tasks, etc), now it seems that I did not misunderstand or missed important results in the first-round reading.

---

> ### Author Response · Authors · 2022-11-30
> **Looking Forward to Your Reply**
>
> Dear Reviewer ndtR ,
>
> Thank you for taking time to read and review our paper. We have revised the text following your suggestions and have provided clarifications on where you may find each claim in the abstract. We have also clarified the novelty and methodological contributions of the paper in the general response.
>
> As the discussion period is nearing the end, we would appreciate if you could kindly check our response. Please don’t hesitate to ask us if you have any more questions.
>
> Thanks,
> Authors

---

### Official Review · Reviewer_5cCF · 2022-10-24

**Confidence:** 4
**Correctness:** 3
**Technical Novelty And Significance:** 2
**Empirical Novelty And Significance:** 3
**Recommendation:** 6

**Clarity, Quality, Novelty And Reproducibility:**

* The paper is well-written in general with some clarity issues that can be addressed.
* The overall quality of this work is good. It presents a interesting frameworks for compositional generation and empirical insights that result in improved performance.
* This work mostly builds on previously explored techniques but brings new insights and improved results.
* Experimental details have been provided in the appendix which aid reproducibility but a code release would have been better.

**Strength And Weaknesses:**

[Strengths]
* The paper presents an interesting study on the use of different samplers and how they significantly change the quality compositional generation.
* The proposal of the energy-based parameterization, although computationally expensive, is well motivated.
* Multiple qualitative and quantitative experiments have been presented that show the improvements offered by the proposed method.
* I appreciate that the authors have clearly discussed the limitations of their proposals.
* The paper is overall well written but there exist some clarity issues (see weaknesses).

[Weaknesses]
* Clarity:
    * For readers unfamiliar with the background, please clearly define what is meant by reverse diffusion.
    * The jump to "improving sampling with MCMC" is abrupt. It is unclear how annealed MCMC improves the results over reverse diffusion. Furthermore, the authors have completely abstracted the details (e.g., the kernel k_t) in this section which forms one of the central theses of this work. I would like to see a more detailed discussion clearly explaining how LMC/HMC helps with compositional generation.
* With the choice of LMC/HMC samplers, I would have liked to see a more detailed discussion on continuous-time diffusion models (e.g., [1]).
* I would have liked to see more examples/experiments of mixture and negation compositions.

[1] Yang Song, Jascha Sohl-Dickstein, Diederik P Kingma, Abhishek Kumar, Stefano Ermon, and Ben Poole. Score-based generative modeling through stochastic differential equations. arXiv preprint arXiv:2011.13456, 2020

**Summary Of The Paper:**

The paper proposes methods to compose diffusion models as products, mixtures, and negations of distributions. The use of LMC and HMC samplers is first motivated by the analysis of the intermediate distributions given by the reverse diffusion. An energy-based parameterisation of diffusion models is presented which enables mixture composition and allows Metropolis corrections in the samplers. Empirical evaluation of compositional generation is provided on multiple datasets and tasks.

LMC: Langevin Monte Carlo,
HMC: Hamiltonian Monte Carlo

**Summary Of The Review:**

Overall, the paper presents an interesting framework for compositional generation. The technical contribution is limited but the paper brings empirical insights that may be relevant to the generative modeling community. There also exist some issues with the clarity which can hopefully be addressed in a revision.

---

> ### Author Response · Authors · 2022-11-19
> **Thank you for your feedback!**
>
> We thank the reviewer for their time and feedback. We address concerns in order:
>
> > Clarity
>
> We have modified the text in Section 2 and 4 to more clearly introduce the reverse process mentioned in the paper and have added more detail on why MCMC corrects the reverse process. We chose not to explicitly mention the kernel $k_t$ because our method works with any MCMC sampler -- the main proposal of our work is that MCMC sampling can be used to sample from a series of intermediate annealed distributions of the composed distribution. It is important to note that this MCMC process we propose is not the same MCMC process used to sample in continuous time diffusion models -- that corresponds to an infinitely discretized reverse process and will lead to incorrect samples from composed distributions.
>
> > Comparison with Continuous Time Diffusion Models
>
> We have clarified differences between models in Section 4 of the paper. As mentioned in the response above, the ULA procedure discussed in continuous time diffusion models is a MCMC procedure over time/distributions and corresponds to an infinitely discretized reverse diffusion process.  As discussed in Section 4 -- the reverse diffusion process over composed distributions is not known, and thus sampling using continuous time diffusion models will lead to incorrect results. In contrast -- we define a discrete series of annealed distributions we wish to sample from to obtain our final composed distribution samples. Our MCMC procedure at each timestep seeks to correctly sample from each specified intermediate annealed distribution.
>
> > Additional Mixture / Negation Samples
>
> We have added additional illustrations of both mixture and negation samples in Appendix F of the paper.

---

> > ### Comment · Reviewer_5cCF · 2022-11-23
> > **Response**
> >
> > Thank you for your response and revision. The clarity has definitely improved in the revision.

---

### Official Review · Reviewer_P5qB · 2022-10-25

**Confidence:** 3
**Correctness:** 3
**Technical Novelty And Significance:** 3
**Empirical Novelty And Significance:** 2
**Recommendation:** 5

**Clarity, Quality, Novelty And Reproducibility:**

Clarity: The authors can improve the writing a bit. I found it a bit hard to parse through the paper.

Quality and novelty: The paper seems to address an important problem. But I am a bit concerned about the similarity of this work with [1] as I mentioned above. The authors could have invested a bit more time in making experiments more rigorous. For instance, comparing with existing papers, other open source methods like GLIDE, stable diffusion, etc.

Reproducibility: The authors mentioned many of the experimental details in the appendix.

**Strength And Weaknesses:**

Strengths:
The idea of composing distributions is an important one. One of the fundamental limitations of the current text to image models is the inability to compose several relationships, hence this problem is an important one to address. The paper takes a principled approach for this problem by considering an energy minimization perspective. They come up with good sampling algorithms that seem to improve over baseline.

Weakness:
One of the main weaknesses I see is the similarity between this paper and [1]. This paper addresses the exact same problem and they take an energy minimization perspective. This paper considers both product distributions and negations.  They use MCMC as well. The authors in this paper cite this work, but they don't mention the similarities and differences between this work. I find this very concerning. I think the difference lies in the samplers used. This paper uses HMC which [1] doesn't. But I'm still unable to assess how this is better than [1].

With respect to experimental validation, the authors show that using HMC improves over other baseline samplers. But, it would have been nice to include comparison with other papers like what [1] did. In fact, [1] had a table for CLEVR dataset. It would have been nice if the authors showed the comparison with that.

[1] Liu et al., "Compositional Visual Generation with Composable Diffusion Models", ECCV 2022

**Summary Of The Paper:**

This papers shows how diffusion models can be combined together to compose several relationships. The authors take an energy minimization perspective of diffusion models. From this perspective, sampling from the distribution can be seen as running an MCMC. Then, the authors show how different distributions can be composed together. They consider three types of distributions - product, mixture and negation. To sample from these distributions, the authors propose annhealed MCMC algorithm, which is explained in appendix. The authors then show some experimental results where their proposed sampler improves composition compared to baselines.

**Summary Of The Review:**

While the paper addresses an important problem, I feel experimental rigor is lacking and the authors could have done a better job comparing with other papers.

---

> ### Author Response · Authors · 2022-11-08
> **Thank you for the feedback!**
>
> We thank the reviewer for their time and thoughtful comments. In your review, you mention similarities with [1]. We agree, our work addresses the same problem, but we tackle it in a different way and build heavily upon that work. In your review you ask for a comparison to [1]. We apologize for not making this very clear in our work, but in the original submission we do compare with [1]. We now understand that this could have been more clearly stated and the updated version of the paper includes a note in the introduction to Section 5 making this much more clear. The main baseline used throughout the work is the score-parameterized diffusion model with reverse diffusion sampling. This is exactly the method of [1] which we compare against in all of our experiments. We can see clearly in Table 1 and Figure 2 that this method has difficulties on even toy problems which are remedied by our proposed improvements (energy-based parameterization and annealed MCMC sampling). As well in Tables 2 and 3 we see, quantitatively, that our method leads to notable improvements over [1].
>
> In terms of qualitative differences between our approach and [1] – by explicitly defining an energy value for sampling, our approach enables us to compose models using the new compositional operator of mixtures – which we demonstrate in Figure 2 and Figure 5. Furthermore, an explicit limitation discussed in [1] is that multiple independently trained diffusion models may not be combined – we illustrate in Figure 2 that our approach directly enables such independently trained models to be composed.
>
> Next you ask for comparisons to existing open source diffusion model implementations. We do not feel this would necessarily improve understanding. Our work was meant to explore the ideas of compositionality within the context of diffusion models generally – abstracted from any 1 diffusion model implementation. The methods and samplers we developed in this work can be applied to any diffusion model implementation. We felt that the most reliable and informative way to demonstrate the impact of our proposed improvements (energy-parameterization + MCMC sampling) was to provide an apples-to-apples comparison with unified architectures, training data, and compute. We do not believe it would be particularly informative to provide a qualitative comparison between, say our energy-parameterized models and the latent-space stable diffusion models as there would be too many differences between the models and training data for us to reliably claim that any improvements observed were solely from our proposed method. Having said that, we could easily apply our method to, say, stable diffusion (although the model would need to be retrained if we wanted to use the energy-based parameterization). This is an exciting idea and we would love to try it out but we currently do not have access to the resources necessary to retrain such a large model.
>
> We hope this addresses your concerns with our work. Thank you.
>
> [1] Liu et al., "Compositional Visual Generation with Composable Diffusion Models", ECCV 2022

---

> ### Author Response · Authors · 2022-11-30
> **Looking Forward to Your Reply**
>
> Dear Reviewer P5qB,
>
> Thank you for taking time to read and review our paper. We have clarified our differences with respect to [1] (we actually compare against [1] across all quantitative results in the paper) and have revised the text to make it more clear. To see more differences with respect to [1], please also see the general response.
>
> As the discussion period is nearing the end, we would appreciate if you could kindly check our response. Please don’t hesitate to ask us if you have any more questions.
>
> Thanks,
> Authors

---

### Official Review · Reviewer_8B2n · 2022-10-28

**Confidence:** 3
**Correctness:** 4
**Technical Novelty And Significance:** 3
**Empirical Novelty And Significance:** 3
**Recommendation:** 6

**Clarity, Quality, Novelty And Reproducibility:**

It is a pleasure to read this paper. It is well-written, and the idea is very clearly illustrated and easy to follow.
The idea is simple, powerful, and effective as illustrated by the paper.
Components of the method exist in the community, but using these techniques together to solve the challenge of composing distributions for image generation is novel.

**Details Of Ethics Concerns:**

No ethics concerns.

**Strength And Weaknesses:**

Strength:
- The paper clearly illustrated the challenge of composing data distributions for image generation and proposed an effective way to solve it.
- The idea is clearly described and the intuition is clear.
- Sufficient experiments to support its effectiveness.

Weakness:
- It might be better to discuss some failure cases for using this approach.

**Summary Of The Paper:**

This paper proposes to use composition operators (product, mixture, negation) to modify the distribution for data generation. To do so, the authors proposed an energy-based parameterization of diffusion models. HMC (Hamiltonian Monte Carlo) is used for sampling rather than the reverse diffusion method. A simple illustrative and quantitative evaluation of the downstream tasks supports this design choice. In the experiment, the method's strength was demonstrated using many image generation tasks.

**Summary Of The Review:**

This paper proposed a simple and effective way to compose distributions for image generation. It is strong in both theory and experiment. The paper is nicely written and easy to follow.

---

> ### Author Response · Authors · 2022-11-09
> **Thanks for your review**
>
> We thank the reviewer for their kind words about our work. We are glad you found it easy to follow and enjoyed reading it. In your review you suggest that the paper would be improved if more discussion is added about failure modes of our approach. We agree completely. We have updated the paper adding to the limitations section discussing cases where model composition is not likely to be successful.
>
> Expanding on that discussion here, let's assume we want to build a product model p_prod(x) = p_1(x)p_2(x)/Z where p_1(x) and p_2(x) are models we have trained on finite sample of data. p_prod(x) will have support at the intersection of p_1(x) and p_2(x). If the region of intersection is far from the region of support of p_1(x) or p_2(x) then, to build an accurate model of p_prod(x) we will need p_1(x) and p_2(x) to be accurate, very far away from the data used to train these models. For example (as mentioned in the paper) if p_1(x) is trained to approximate N(-10, 1) and p_2(x) is trained to approximate N(10, 1), then they will both need to be accurate near the origin. But, the likelihood we will observe any data around the origin while training p_1 and p_2 is near 0. The same can be said for text-2-image composition with highly contradictory prompts such as p(x | "a bright day") and p(x | "a dark night").
>
> We hope this discussion and the changes to the paper address your concerns. Please let us know if there is anything else we can address to improve your opinion of our work. Thank you.

---

> > ### Author Response · Authors · 2022-11-16
> > **Added new section**
> >
> > Hello!
> >
> > We would like to inform you that we have added a new section to the Appendix, section E which presents more 2D composition results and demonstrates failure cases as discussed in our previously added section.

---

### Author Response · Authors · 2022-11-08
**Thanks for the feedback!**

We thank all of the reviewers for their thoughtful and helpful feedback. We are in the process of responding to each of you and incorporating your feedback into the paper. We will keep track of changes to the paper in this thread and respond to each of you individually addressing your comments and when we have incorporated your individual feedback into the paper.

We hope that our responses and changes to the paper address your concerns about our work.

--- Changes to the paper ---

Addressing reviewer P5qB's comments about comparisons to [1], we have added a note to the end of Section 5's introduction explaining that the baseline method of a score-parameterized diffusion model using reverse diffusion sampling is identical to the method of [1]. We compared against this baseline in all of our experiments and apologize for not making this more clear.


[1] Liu et al., "Compositional Visual Generation with Composable Diffusion Models", ECCV 2022

---

> ### Author Response · Authors · 2022-11-09
> **More changes**
>
> In response to reviewer 8B2n's feedback, we have expanded the limitations section (section 6) to include a discussion of situations where model composition may fail.

---

> > ### Author Response · Authors · 2022-11-16
> > **Even more changes**
> >
> > In response to reviewers 8B2n and ndtR asking for further discussion on failure cases we have added Appendix section F which expands on our previously added discussion of failure cases. We add results on 2D datasets and visualize cases where composition fails. We also add results on 2D datasets for negation.

---

### Author Response · Authors · 2022-11-20
**General Response to All Reviewers**

We thank all reviewers for their detailed feedback and attention to the paper. Here, we summarize our overall rebuttal response to reviewers as well as the additional experiments we have run. All reviewers appreciated the underlying motivation / problem tackled by the paper, but there are some concerns of novelty which we discuss below.  Please see the individual reviewer responses for more detailed discussion to individual reviewer concerns.

____________
# Novelty
____________

In this paper, we aim to study the problem of how diffusion models may be combined both with each other as well as with other models. This problem setting may be useful for a variety of different downstream tasks (for instance highly structured conditional generative modeling) and is relatively under-explored, with the most relevant work, as noted by Reviewer P5qB, being [1].

In [1], multiple instances of the **same trained diffusion model** are composed together for compositional visual generation. To compose separate diffusion models together, the authors propose to combine their score functions (which corresponds to the reverse diffusion baseline that we compare with in all experiments in the paper).  As noted in the limitations section of [1], the authors were unable to use this approach to compose **separately trained diffusion models together**.

In this paper, we illustrate in the beginning of Section 4 and in Appendix D that mathematically combining score functions together to sample from the diffusion process is incorrect.  Instead, it is necessary to define a series of intermediate annealed composed distributions, which we may sample from using MCMC. With these mathematical fixes, we illustrate in both Figure 1 and Table 2 how **separately trained diffusion models may be correctly composed together**. Furthermore, different from [1], we explore how diffusion models may also be composed with other models such as classifiers and illustrate that the most common classifier-based guidance approach used with diffusion models is mathematically incorrect (Appendix D). In addition, we propose an approach which enables different diffusion models to be *“mixed”* together.

We further believe that our paper has several different methodological contributions which are new and may further be of broader interest to general diffusion practitioners. First, our paper motivates the use of energy-based diffusion models, which unlike the traditional score-based parameterization, assigns an unnormalized density to each point sampled along distributions in the diffusion process. We explore in detail different possible parameterization of the energy function in Appendix E, and illustrate that energy-based parameterization both enables Metropolis adjusted MCMC procedures as well as mixture compositions between separately trained diffusion models. We further illustrate that energy-based parameterizations may be successfully scaled to complex generative modeling tasks, generating text-to-image samples with high visual quality (Figure 5, 6, A1, A2, A3, A4). In contrast, [2] explore a single energy parameterization of diffusion models on only the CIFAR-10 dataset and come up with a final conclusion that there is no reason to use an energy parameterization in diffusion models.

Second, our paper proposes the use of more sophisticated  MCMC samplers in diffusion models. Our work proposes the use of Metropolis adjusted MCMC sampling to generate samples from diffusion models. We illustrate in Figure 3b) how Metropolis adjustment substantially improves sampling performance across timesteps over unadjusted variants (which are the de facto standard when sampling from diffusion models). We further illustrate how integrating momentum into sampling (using HMC sampling) can further substantially improve generations from diffusion models. Neither of these techniques have been integrated in existing diffusion sampling procedures and this challenges the de facto belief that unadjusted Langevin is sufficient to sample from diffusion models.


[1] Liu et al., "Compositional Visual Generation with Composable Diffusion Models", ECCV 2022

[2] Ho and Salimans, "Should EBMs model the energy or the score?", ICLR 2021 EBM Workshop

--------------
# Additional Experiments
--------------

**Additional Examples of Negations and Mixtures** We have added additional qualitative examples of our approach when negating and mixing different 2D distributions in Appendix Figure A5 and A6 following suggestions from Reviewer 5cCF.

**Failure Cases of our Method** We have illustrated some  failure cases of our model when composing distributions with disjoint support in Figure A7 of the appendix following suggestions from Reviewer ndtR and 8B2n and have also discussed limitations of our approach in the conclusion of the paper.

---

### Decision · Program_Chairs · 2023-01-20

**Decision:**

Reject

**Justification For Why Not Higher Score:**

The reviewers remained largely unenthusiastic about this paper, despite it being technically good and well written. Fundamentally, I think it was considered too incremental.

**Justification For Why Not Lower Score:**

N/A

**Metareview: Summary, Strengths And Weaknesses:**

The paper describes an energy-based diffusion models for compositional generation. The paper is well written and the ideas are clearly explained. The weakest part according to the reviews is the novelty, given similar approaches to composing diffusion models. This is a solid piece of work and might inform the community of the utility of these models. However, I feel that some of the points are not particularly well motivated. For example why should "AND" correspond exactly to taking the product of two distributions p1(x)p2(x) -- there are other ways to do this (for example p1(x)^alpha*p2(x)^beta). Is there a principle way to learn what the "best" way to combine distributions is?  On the whole this is a solid contribution, but the reviewer scores remained borderline after the discussion.

**Summary Of Ac-Reviewer Meeting:**

This paper was on the lower limit of  borderline.